# Precipitation extremes in Ukraine from 1979 to 2019: Climatology, large-scale flow conditions, and moisture sources

Ellina Agayar[1,2], Franziska Aemisegger[1], Moshe Armon[1], Alexander Scherrmann[1], and Heini Wernli[1]

[1]Institute for Atmospheric and Climate Science, ETH Zürich, Zürich, Switzerland
[2]Odesa State Environmental University, Odesa, Ukraine

*Correspondence to*: Ellina Agayar (ellina.agayar@env.ethz.ch)

**Abstract.** Understanding extreme precipitation events (EPEs) and their underlying dynamical processes and moisture transport patterns is essential to mitigate EPE-related risks. In this study, we investigate the dynamics of 82 EPEs ($\geq 100$ mm·day$^{-1}$) over the territory of Ukraine in the recent decades (1979-2019), of which the majority occurred in summer. The EPEs are identified based on precipitation observations from 215 meteorological stations and posts in Ukraine. The atmospheric variables for the case study analysis of selected EPEs and for climatological composites and trajectory calculations were taken from ERA5 reanalyses. Moisture sources contributing to the EPEs in Ukraine are identified with kinematic backward trajectories and the subsequent application of a moisture source identification scheme based on the humidity mass budget along these trajectories. The large-scale atmospheric circulation associated with EPEs was studied for a selection of representative EPEs in all seasons and with the aid of composites of all events per season. Results show that EPEs in summer occur all across Ukraine, but in other seasons EPE hotspots are mainly in the Carpathians and along the Black and Azov Seas. All EPEs were associated with a surface cyclone, and most with an upper-level trough, except for the winter events that occurred in situations with very strong westerly jets. Isentropic potential vorticity anomalies associated with EPEs in Ukraine show clear dipole structures in all seasons, however, interestingly with a different orientation of these anomaly dipoles between seasons. The analysis of moisture sources revealed a very strong case-to-case variability and often a combination of local and remote sources. Oceanic sources dominate in winter, but land evapotranspiration accounts for 60-80% of the moisture that rains out in EPEs in the other seasons. Taken together, these findings provide novel insight into large-scale characteristics of EPE in Ukraine, in a region with a unique geographical setting and with moisture sources as diverse as Newfoundland, the Azores, the Caspian Sea, and the Arctic Ocean.

**Keywords:** extreme precipitation events, Ukraine, potential vorticity anomalies, large-scale circulation, moisture sources

## 1 Introduction

Anthropogenic climate change not only affects mean climate conditions, but changes are also expected in the temporal variability of extreme meteorological events, including precipitation. Extreme precipitation events (EPEs) can lead to severe socioeconomic impacts and are expected to change in severity, frequency, and duration because of anthropogenic global warming (IPCC, 2021). EPEs pose a great threat as a trigger for landslides and floods (Jonkman, 2005; Barton et al., 2016; Jonkeren et al., 2014; Madsen et al., 2014; Moore et al., 2020). They are one of the most frequent natural hazards as documented for many regions of the world (Winschall et al., 2014, Santos et al., 2016; Li and Wang, 2018; Mastrantonas et al. 2020; Mastrantonas et al., 2021; Gao and Mathur, 2021; Giuntoli et al., 2021; Armon et al., 2023), and Ukraine is not an exception.

Ukraine is characterized by a quite complex orography. In the west and south-east are mountain ranges the Carpathians (Hoverla) and the Crimean Mountains (Roman-Kosh), with maximum elevations of 2061 m and 1545 m, respectively. In the south are the Black and Azov Seas, and most of the territory is characterized by hills (with typical heights of 200-300 m) and low-land plains. The extended geographical domain covered by the country includes a variety of climatic zones, e.g., the climate of the mountain tundra in the Carpathians and the Crimean Mountains, and subtropical climate along the southern coast of Crimea. Effects of continentality increase from west to east. Maritime air frequently passes over Ukraine from the North Atlantic, the Mediterranean, and the Arctic seas. In periods without advection of maritime air, continental conditions prevail with air circulating over the Eurasia plains (Lipinskyi et al., 2003). Recent studies already documented ongoing

climatic changes in Ukraine using observations and numerical model simulations (e.g., Semerhei-Chumachenko et al., 2020; Martazinova et al., 2018; Osadchy et al., 2012). These changes also lead to a dramatic increase in average annual economic losses due to flooding. An example is the flood in Transcarpathia in the period 21-27 June 2008, and a catastrophic flood in summer 2020, when in five regions in the west of the country, floods affected at least 250 settlements, damaged 750 km of roads, and 4 people died (Ukrainian State Agency of Water Resources,2020; Mykhailiuk , 2022).

The genesis and spatiotemporal variability of EPEs in midlatitude regions are a consequence of complex dynamical and thermodynamical processes that occur on the synoptic and mesoscale. The nature of these processes is determined both by the large-scale atmospheric flow, leading to a strong increase in moisture transport to the EPE region, and the influence of deep convective systems. For instance, short-term EPEs usually are a consequence of intense convection. In contrast, EPEs accumulated over 1–3 days are often associated with the passage of an atmospheric front (Catto and Pfahl, 2013), with upper-level Rossby wave breaking (Massacand et al., 1998; Moore et al., 2019; de Vries, 2021), and also cyclones and blocking systems were shown to be especially relevant for EPEs (Pfahl 2014; Priestley et al., 2017; Agel et al., 2018; Tuel et al., 2022). It is quite common for heavy precipitation to occur in synoptic configurations at the interface between high-pressure disturbances and cyclones (Breugem et al., 2020). According to Pfahl and Wernli (2012), in many regions, cyclones are linked with a large percentage of EPEs. Cyclones and anticyclones both play an important role in moisture transport, while cyclones typically also go along with forcing for ascent, in combination leading to EPEs. Blocking anticyclones in addition effectively hinder the usual westerly large-scale atmospheric flow, resulting in persistent flow anomalies in and around the blocked region. Their presence and characteristics significantly impact the predictability of weather extremes (Rex, 1950a; Lenggenhager et al., 2019; Kautz et al., 2022), including EPEs. Furthermore, extreme precipitation is often associated with atmospheric blocking and coexisting upper-tropospheric cutoffs (Portmann et al., 2021). A key aspect of EPEs that gained increased attention in the last years, is the analysis of moisture sources. For instance, James and Stohl (2004) and Sodemann et al. (2008) developed trajectory-based methods to objectively identify evaporative regions that later contribute to intense rainfall in the region of the EPE. Such methods have been applied to identify the moisture sources globally (Gimeno et al., 2012; Sodemann, 2020; for selected EPEs in Europe (e.g., Grams et al. 2014; Raveh-Rubin and Wernli, 2017) for climatological analyses of precipitation in the Alpine region (Sodemann and Zubler 2009), the Mediterranean (Ciric et al., 2018), the USA (e.g., Yang et al., 2023) in South Asia (Bohlinger et al., 2017), and in the Arabian Peninsula (Horan et al., 2023). However there has not been much research on the hydrological cycle in Ukraine since the study by Budyko and Drozdov (1953), and moisture sources for EPEs in this domain have not been investigated yet. In this study, we consider precipitation observations to identify EPEs for the territory of Ukraine in the last 40 years and study their characteristics in terms of the large-scale flow and moisture source conditions. For this, we use the ERA5dataset, which is the fifth-generation reanalysis from the European Centre for Medium-range Weather Forecasts (ECMWF) that is available since 1940 (Hersbach et al., 2020). ERA5 provides hourly estimates for a large number of atmospheric, ocean-wave and land-surface quantities. The novelty is in the application of a systematic climatological approach to study the large-scale characteristics of EPEs in Ukraine and their moisture sources. Given the geographical setting of Ukraine, with its proximity to the Black Sea, the eastern Mediterranean, but also the Baltic and the Caspian Seas, the most important moisture sources are not obvious and require careful analysis. Using the ERA5 dataset, anomalous characteristics of the flow situation associated with EPEs can be identified, including potential vorticity (PV) and wind speed at different levels, in all seasons. More specifically, this study is guided by the following key questions:

• 1: What is the seasonality of EPEs in Ukraine, and how are EPEs distributed spatially and temporally?

• 2: What are the distinctive tropospheric flow conditions during EPEs?

• 3: What are the geographical moisture sources of EPEs in Ukraine in the different seasons?

• 4: What is the distinction between individual cases with diverse large-scale flow conditions and moisture source origins?

The paper is organized as follows. In Sect. 2 we introduce the datasets and methods. Then, in Sect. 3, a climatological

overview of EPEs is presented (Sect. 3.1). In Sect. 3.2, we discuss anomalies of pressure and summer moisture anomalies of

EPEs, in Sect. 3.3 we analyze PV and wind anomalies associated with EPEs. Seasonal moisture source identification

(Sect. 3.4) and selected case studies of EPEs illustrate the main large-scale processes involved (Sect. 3.5). A summary and

conclusions are given in Sect. 4.

**2 Data and methodology**

**2.1 Identification of EPEs**

For this study, 215 meteorological stations and posts (including aviation weather stations, gauging stations, etc.) with daily

data from 1979 to 2019 are used. From this dataset, 183 stations were selected for our study that have a complete set of data.

The remaining 32 stations did not have the same record length for various reasons. Nevertheless, these stations were still

tested for the occurrence of EPEs, but no extreme events were found according to our criteria (see below). Due to the

absence of data in the Ukrainian meteorological network for certain regions of Crimea from February 2015 to December

2019, additional data were obtained using open-access observations for this region (SYNOP observational data).

Unfortunately, data for four stations in the Donetsk and Lugansk regions for the period of 2015-2019 are not openly

available. In this region, a 36-year dataset was employed to identify days with extreme precipitation.

Our criterion to identify EPEs was a threshold of 100 mm·day$^{-1}$. With this criterion, in total 82 EPEs were identified.

Table S1 in the Supplement lists the date and station for each of these events. Our threshold of 100 mm day$^{-1}$, is chosen from

expert knowledge, as it is often used to define EPEs in different countries. For instance, Martin-Vide et al. (2008) used this

threshold to determine EPEs in the western Mediterranean, and Tramblay et al. (2013) in southern France. Boissier and Vinet

(2009) identified the value of 100 mm·day$^{-1}$ as a critical threshold that could trigger fatalities. Also in Ukraine, this threshold

is used to identify an event as extreme. Given that we considered a 40 year time period and that EPEs were identified at each

station between 0 and 3 times (see Table S1), we can estimate that our threshold corresponds to the 99.8th percentile or

higher. These percentiles highlight that the selected threshold of $\geq$ 100 mm·day$^{-1}$ indeed selects extreme, i.e., very rare events.

These events are so rare that we cannot robustly assess regional differences of percentiles. The largest amount of recorded

precipitation occurred with 278 mm·day$^{-1}$ on 2 Sep 1981 in Karadag (in the southeast of Crimea). Another exceptional event

occurred in Ai-Petry (in the south of Crimea) with 228 mm on 27 and 28 Dec 1999 (accumulated over two days). An

overview of the seasonal and geographical distribution of the EPEs will be given in Sect. 3.1.

**2.2 Dynamical Characterization**

For the dynamical investigation of the EPEs, selected fields from ERA5 reanalyses from the ECMWF were used. All

reanalysis data were interpolated to a 0.5° grid. Specifically, we analyzed the following variables, characterizing the large-

scale flow: mean sea level pressure (MSLP), wind speed at 300 hPa , geopotential height at 500 hPa, PV on different

isentropic surfaces, total precipitation, total column water (TCW) and convective available potential energy (CAPE).

Composites were calculated as the mean of all values during EPE days and anomalies were computed as deviations between

the seasonal average and the mean EPE conditions. To overcome biases related to intra-season differences in the number of

identified EPEs (e.g., no events occur in January and April), we corrected the above-mentioned composite anomalies with

this frequency bias of EPEs serving as a weight for the seasonal averaging. Consideration of such standardized anomalies for EPE events can help recognize typical flow conditions and potential precursors for EPEs in Ukraine.

**2.3 Moisture Sources**

We used the Lagrangian Analysis Tool LAGRANTO (Sprenger and Wernli, 2015) and 3-dimensional wind fields from ERA5 to compute 10-day backward trajectories from the regions affected by EPEs. For the identification of moisture sources, we used the method introduced by Sodemann et al. (2008), which relies on the evolution of specific humidity along the trajectories. An analogous trajectory-based approach has been used previously for identifying moisture sources of precipitation in, e.g., intense North Atlantic cyclones (Aemisegger, 2018; Papritz et al., 2021), Mediterranean cyclones (Krug et al., 2022), and for a climatological analysis of the global water cycle (Sodemann, 2020). We started the trajectories every hour on the day of the EPE and every 20 hPa between 1000 and 200 hPa from the location of the station, where the EPE occurred. Trajectories were considered for the moisture source diagnostic if their relative humidity at the arrival point exceeded 80%. Since the global mean atmospheric moisture residence time is about 4–5 days (Läderach and Sodemann, 2016) the 10-day backward trajectories cover a large part of the moisture sources of the total precipitation, with explained fractions of 85 to 97%. Moisture source regions were identified by diagnosing hourly changes in specific humidity along the air parcel trajectories, and assuming that increases in specific humidity result from surface evaporation and decreases from precipitation. Evaporation is identified where the hourly increase of specific humidity exceeds $0.025$ g·kg$^{-1}$·h$^{-1}$. These moisture uptakes were taken into account both in and above the boundary layer, since convective injections of vapour from the boundary layer can also occur in the free troposphere (Aemisegger et al., 2014). When precipitation occurs (identified as decreases in specific humidity), the contribution of the previous uptakes are discounted proportionally to their share in the humidity loss (Sodemann et al., 2008). This moisture source diagnostic was applied to the hourly trajectories for all EPE days, and moisture uptake maps were calculated for each EPE. For the climatological analysis four seasonal composite maps were calculated by weighting each event by its total measured precipitation.

**3 Results**

**3.1 Spatiotemporal distribution of EPEs over Ukraine**

Precipitation in Ukraine generally exhibits a diminishing trend from the north and northwest to the south and southeast areas (Lipinskyi et al., 2011). In the mountainous areas, orographic lifting contributes to enhanced precipitation. As a result, the Ukrainian Carpathians and the Crimean Mountains experience the largest precipitation values (annual total >1000 mm). In the central and eastern parts of Ukraine, the amount of annual precipitation is 550 - 650 mm; the southern part, along the coast of the Black Sea, is comparatively dry (annual total 380 – 400 mm). In the cold season, approximately 20–25% of the annual precipitation occurs, contrasting with the warm period, where 75–80% of the total annual precipitation is recorded. During the warmer season, the precipitation distribution reflects the annual pattern, with a gradual decrease from the northwest to the southeast, reaching 300 mm or less in the coastal regions.

In the study period, the 82 EPEs identified with a threshold of 100 mm day$^{-1}$, were observed at stations in almost all regions of Ukraine, except Sumy, Luhansk and Cherkasy (Fig. 1). Their distribution has a clear seasonality. The highest number of EPEs occurred in summer (June, July, August) with 54 cases, with a peak in July with 27 cases (Fig. 2a). In the Northern Black Sea region and on the Crimean Peninsula, 18 summer events were observed, two of them with more than

130 mm·day$^{-1}$. For example, in the Odesa region at Serbka station, 148.4 mm·day$^{-1}$ were recorded on 27 June 1996, and at the station Pochtove in Crimea, 137.8 mm·day$^{-1}$ on 23 July 2002. Several summer EPEs were also noted in the central, western, and eastern regions of Ukraine. The most intense precipitation in these areas was recorded at the Loshkarevka station in the Dnepropetrovsk region with 154.2 mm·day$^{-1}$ on 5 July 1983 and at the Barishevka station in the Kyiv region

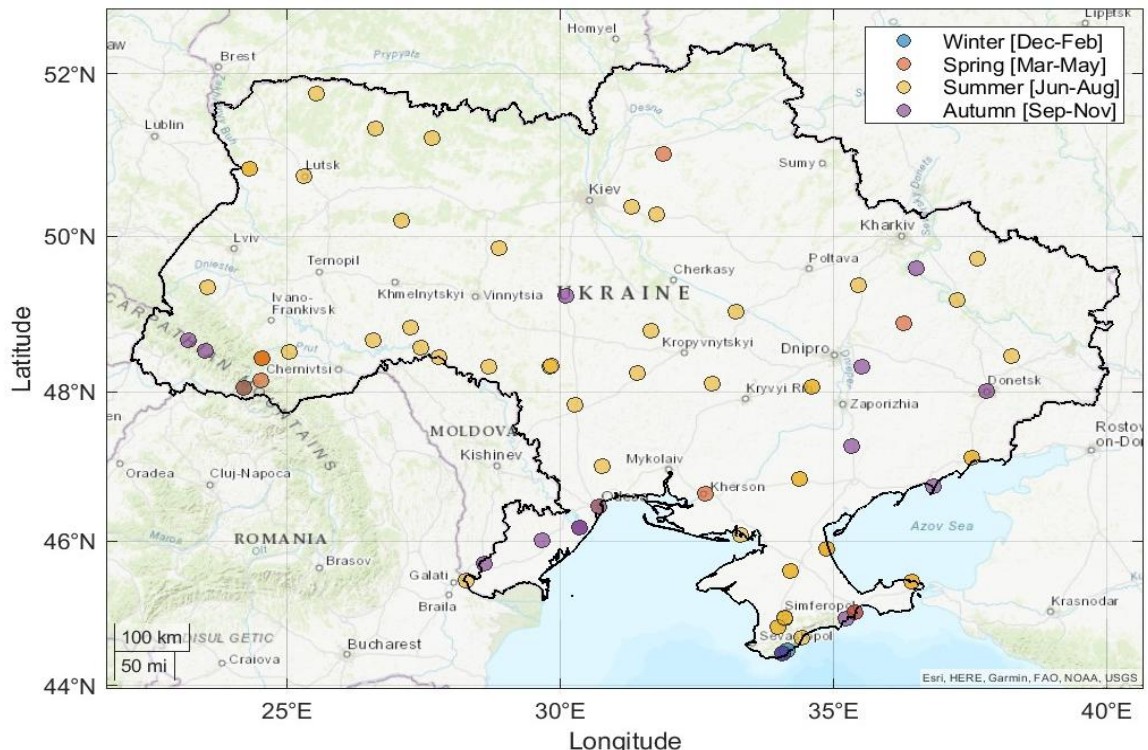

**Figure 1 The identified 82 EPEs at stations in Ukraine in the period 1979-2019. Colors show the season of occurrence. Please note some of the stations recorded more than one EPE. Ukraine shapefile source: https://gadm.org/maps/UKR.html**

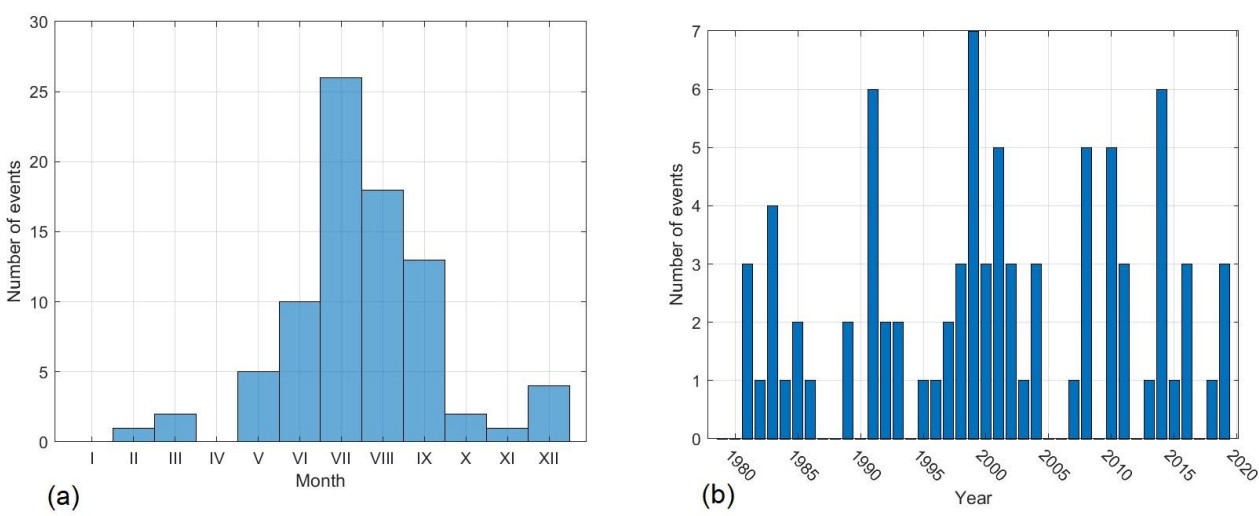

**Figure 2 (a) Seasonal cycle (sum of monthly occurrence values) and (b) time series of annual number of EPEs in Ukraine in 1979-2019.**

with 130.7 mm·day$^{-1}$ on 1 July 2011. In autumn, 16 cases of extreme precipitation were recorded, mainly in the west and south of Ukraine (meteorological stations located in the Transcarpathia and Odesa regions, as well as in Crimea). All autumn EPEs occurred in September, except for three cases (12 Oct 2016 - Bolgrad, 29 Oct 1992 - Play, and 4 Nov 1998 - Mizhirgya). One EPE was noted on the territory of the Crimean Peninsula at Karadag on 1 Sep 1991 with 278 mm·day$^{-1}$. This particularly high value most likely reveals a strong orographic effect on the intensity of EPEs in this region. At Belgorod-Dnestrovsky station (the Black Sea coast), two EPEs were recorded during the study period, both in September, on 21 Sep 2008 (100.2 mm·day$^{-1}$) and on 20 Sep 2016 (135.2 mm·day$^{-1}$). In spring, seven EPEs were identified in Ukraine, mainly in the Ivano-Frankivsk and Zakarpattia regions, as well as in the Crimea. The intensity of precipitation in spring did not exceed 116.9 mm·day$^{-1}$ (on 5 March 2001, station Pozhezhevskaya). The lowest occurrence of EPEs was in the winter, with only five events that all occurred in the south of Crimea, four of them in December and one in February. At the mountain station Ai-Petri 112.5 and 115.3 mm·day$^{-1}$ were observed on 27-28 Dec 1999, and at Yalta 100 mm·day$^{-1}$ were measured on 28 Dec 1999.

The number of EPEs in Ukraine varies from year to year. Actually, they were registered annually, with the exception of a few years (Fig. 2b). In 1991, 1999, and 2014 the number of EPEs rose to 6-7 cases. However, there is no obvious trend in the frequency of EPEs that are identified with the threshold of 100 mm·day$^{-1}$.

**3.2 Dynamical characterization: seasonal-mean flow composites**

In this section, we analyze the dynamic conditions for the occurrence of EPEs in Ukraine, separately for each season. Figure 3 presents the composites of the anomalies of MSLP, geopotential height and the horizontal wind at 500 hPa on EPE days. We first discuss these flow anomalies for all four seasons, and then, put a focus on summer, when most EPEs occurred, where we consider anomalies of total precipitation, CAPE and TCW in Fig. 4. (For completeness, the anomaly maps of these fields for the other seasons can be found in the supplementary Fig. S1).

On winter EPE days (Fig. 3a), there is a strong negative geopotential height anomaly over Eastern Europe, with a peak value of 241 m. This upper-level trough located above a baroclinic zone causes the formation of a cyclone in the lower troposphere. The center of the negative MSLP anomaly is located over the Carpathian region (Hungary and Romania) and reaches values up to 24 hPa below average. A second local MSLP anomaly is found over eastern Ukraine (–20 hPa). The strong cyclones over southwest Europe on EPE days go along with a strongly intensified jet stream over the northern Mediterranean, Turkey, and the Black Sea (maximum 500 hPa wind speed anomalies up to 20 m·s$^{-1}$). The intense low-pressure systems developing over west and southwest Ukraine led to EPEs in the southwestern Ukraine, particularly in Transcarpathia region and on the coast of Crimea, where frontal precipitation is reinforced by orographic uplift on the windward sides of mountain ranges.

In all other seasons, 500-hPa geopotential height anomalies on EPE days are much weaker than in winter, but negative anomalies extend over Ukraine in all seasons. In spring (Fig. 3b), the negative anomaly reaches a maximum amplitude of -48 m. The low-pressure zone covers the entire territory of Ukraine (MSLP anomaly of -6 hPa) and weak 500-hPa wind anomalies curve cyclonically over the Balkans toward the Black Sea. These synoptic conditions led to the emergence of EPEs mainly in the southern regions of Ukraine and Crimea. In autumn (Fig. 3d), a negative 500-hPa geopotential height anomaly is identified over the entire territory of Ukraine, which is strongest in the southeast with peak values of –79 m. At the same time, there is a strong negative MSLP anomaly (up to –12 hPa) with its core located over eastern Ukraine. The wind anomaly in the middle troposphere shows again a cyclonic flow, in agreement with the negative geopotential height anomaly.

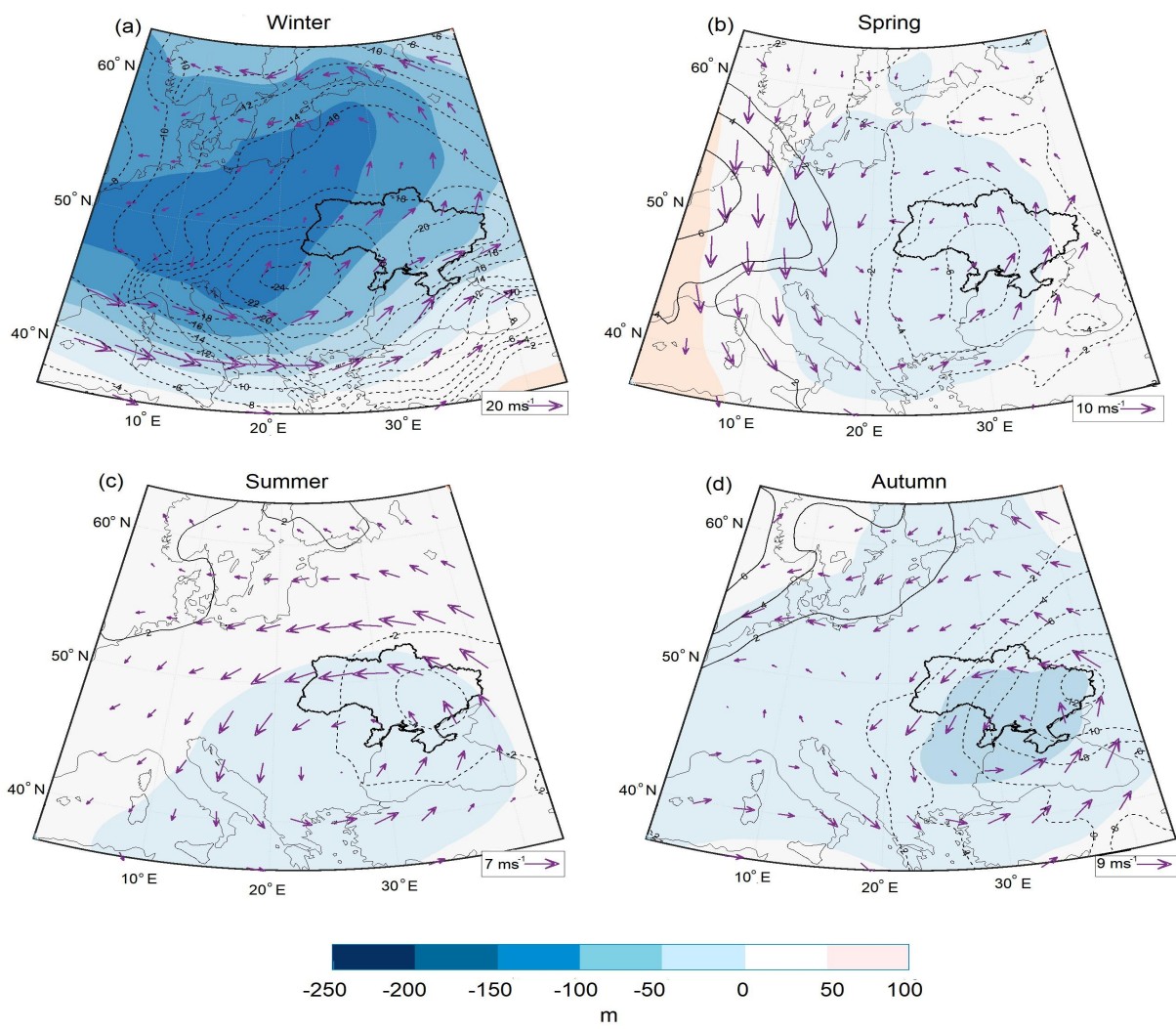

**Figure 3** **Seasonal composites on EPE days of anomalies of geopotential height at 500 hPa (colors, in m), MLSP (in hPa, solid and dashed contours for positive and negative values, respectively), and 500-hPa winds (purple arrows, reference vector is shown in lower right corner, in m·s⁻¹).**

Last but not least, in summer, when most EPEs occurred, surface pressure and 500-hPa level anomalies revealed a weak-gradient depression with a low-pressure center shifted to the southeast of Ukraine (-4 hPa). The negative 500-hPa level anomaly stretched from the central Mediterranean through the Balkans, reaching peak values of –43 m. Wind anomalies at the 500-hPa level over Ukraine were primarily from the east; however, they were weaker compared to those in other seasons and did not exceed 8 m·s⁻¹ (Fig. 3c). EPEs were observed throughout the domain under these large-scale flow structures, but were due to different reasons. EPE in the central and southeast parts coincided with the low anomaly center, indicating that their main cause was dynamical lifting. However, small-scale processes associated with strong convection also played a significant role, as well as orographic ascent along the windward slopes of the Crimean and Carpathian Mountains. The thermodynamic composites shown in Fig. 4 for the summer EPEs provide valuable additional information. Figure 4a reveals that days with an EPE occurring at one station had on average anomalously high precipitation in the entire territory of Ukraine and beyond.  Precipitation anomalies exceed 0.6 mm·h⁻¹ along a band from north to south, stretching through the Podolsk Upland to the Black Sea. Other local maxima were observed in the southeast, including the Donetsk Ridge, the

Azov Sea, and Crimea. Positive total column water anomalies (up to more than 4 kg m$^{-2}$) extend over all of Ukraine, except
the Transcarpathian region (Fig. 4b).

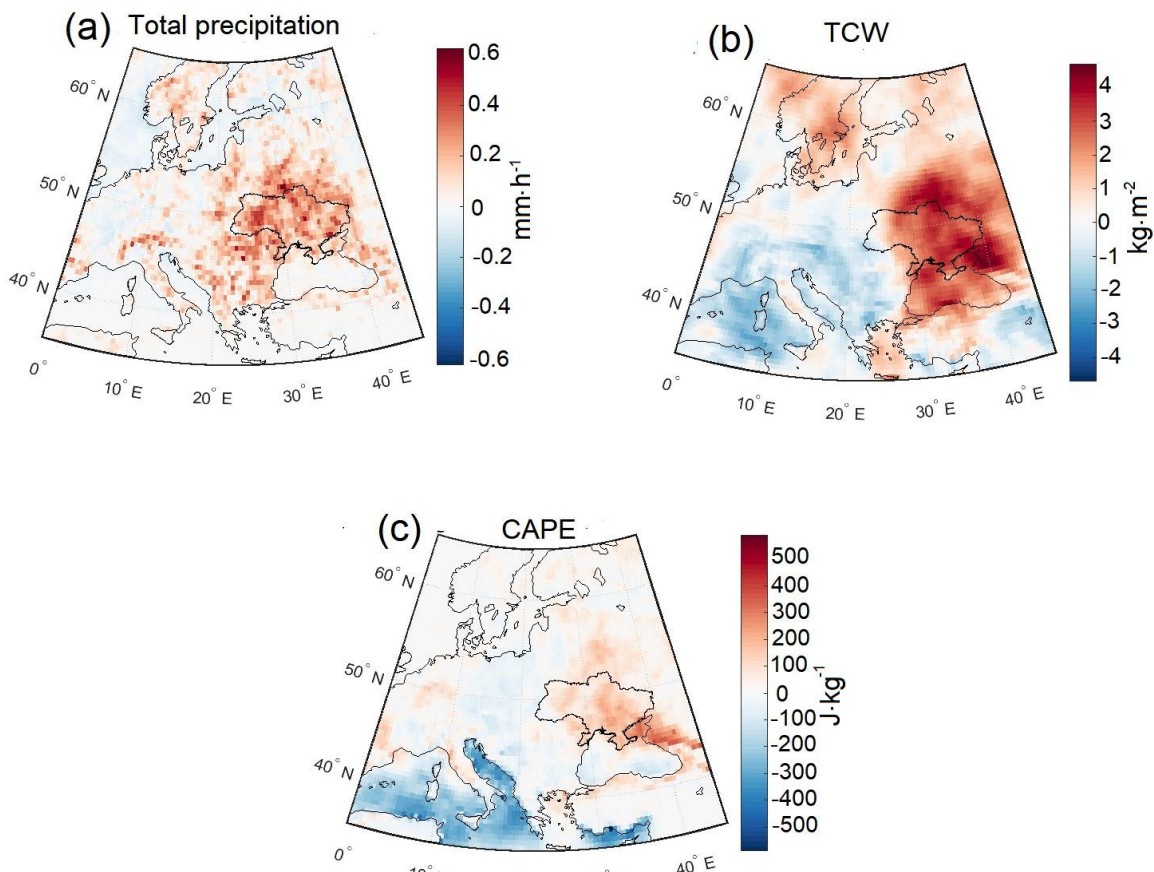

**Figure 4  Anomalies of total precipitation, TCW and CAPE at 15 UTC on EPE days in summer.**
This also contributes to increased CAPE (Fig. 4c), providing the necessary ingredients for convection (Rasmussen and
Houze 2016). Thus, in summer, on days when EPEs occurred, elevated moisture levels progressed over the entire Ukraine,
making the entire domain on average more humid than compared to normal conditions. The EPEs in Ukraine were triggered
both by the large-scale ascent due to the upper-level cyclonic flow anomaly and the development of convection in the
southeastern and eastern regions, as well as by local (orographic) convection in the central and western parts.
In summary, the primary reasons for the occurrence of extreme precipitation in all seasons were the presence of cyclonic
anomalies generating anomalously moist flows and the triggering of convection in the affected regions. In summer, the
greatest contribution to the formation of EPEs was from convective processes, both frontal and local. Total column water
values were, on average, increased, mainly over the eastern regions in winter and autumn, and over the entire Ukraine in
spring and summer (Figs. 4b and S1). This suggests that moisture characteristics are essential for understanding the process
of extreme precipitation formation. Therefore Sect. 3.4 is dedicated to defining the origin, uptake characteristics, and
transport pathways of moisture that precipitates during EPEs in Ukraine. It is also worth highlighting the influence of
orography on the formation of EPEs. Whereas over the flat terrain of Ukraine, dynamic uplift near the cyclone center is most
important for the generation of extreme precipitation, in the mountainous regions of Ukraine, such as the Carpathians and
Crimea, orographic enhancement of precipitation is crucial in the formation of the EPEs.

**3.3 Climatological characteristics of PV anomalies**

In all seasons the vertical coherence of the negative anomalies of geopotential height at 500 hPa and MSLP, which indicates that EPEs typically co-occur with vertically deep extratropical cyclones that are associated with upper-level troughs or cutoffs. This aspect can be further investigated by also considering the isentropic PV distribution on EPE days. Positive PV anomalies on tropopause-intersecting isentropes are often linked with developing surface cyclones and severe weather phenomena (Portmann et al., 2021), a different isentropic surface is most suitable to study upper-level PV dynamics in a specific region during the different seasons. In this study, it turned out to be useful to select the following isentropes: 315 K in winter, 325 K in spring, 335 K in summer, and 330 K in autumn. Figure 5 presents the composites of PV anomalies on these isentropes and the 300-hPa wind (Fig. 5).

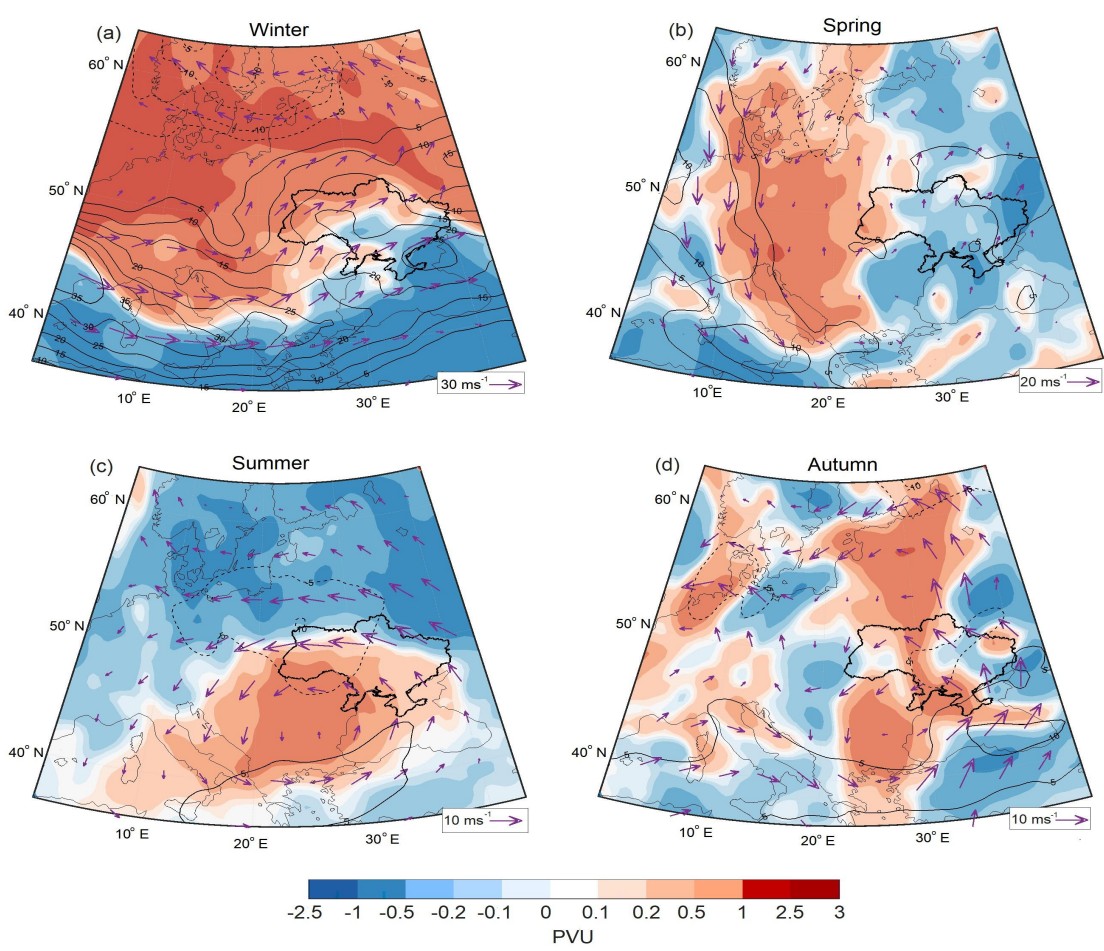

**Figure 5  Seasonal composites on EPE days of anomalies of isentropic PV (colors, in PVU), and of 300-hPa wind (purple arrows, see reference vector in lower right corner, in m·s⁻¹) and wind speed (solid and dashed contours for positive and negative values, respectively).**

On winter EPE days (Fig. 5a), there is a large positive PV anomaly extending over Europe (consistent with the negative Z500 anomaly in Fig. 3a) with maximum values that exceed 2.5 PVU near 55°N stretching from the North Sea to southern Russia. The flow induced by this PV anomaly leads to strong low-tropospheric winds toward the Crimean and Carpathian Mountains causing orographic uplift (not shown). In the south of the depicted region, i.e., over most of the Mediterranean and the Black Sea, there are large negative PV anomalies, and as a consequence, over Ukraine, PV anomalies are close to zero but there is a strong poleward gradient of PV anomalies, which goes along with strongly increased westerly winds at the

300-hPa jet level (wind speed anomalies reached ~25-35 m·s$^{-1}$). Noteworthy is the strongly positive upper-level wind speed anomaly over the Crimea Peninsula and the Azov Sea, i.e., in the region where the most intense winter EPEs have been recorded.

In spring (Fig. 5b), the positive PV anomaly is more confined and extends meridionally from southern Scandinavia to the Adriatic Sea, with maximum values over the eastern Alps (+0.1-1 PVU). This positive anomaly also reaches western Ukraine, but over the main territory of Ukraine, PV anomalies are weakly negative and increase in amplitude towards the east. A strong positive wind speed anomaly occurs along the western flank of the PV anomaly, and over Ukraine there is a weakly enhanced southerly flow at 300 hPa. Similarly to winter, the EPE regions were located east of the positive upper-level PV anomaly, in a region with an enhanced horizontal PV gradient and therefore upper-level flow.

In summer (Fig. 5c), the moderately intense positive PV anomaly is located over southeastern Europe, extending over most parts of Ukraine. In this season, negative PV anomalies occur at high latitudes, leading to a strongly different PV anomaly pattern compared to the other seasons. The summer PV anomaly over Eastern Europe reflects the occurrence of PV cutoffs, which repeatedly formed, locally changing the static stability, and thus providing the ideal mesoscale environment for the triggering of convection and EPEs, and the formation of cyclones over Ukraine. Also note, that the Black Sea region is characterized by a local maximum in the frequency of PV cutoffs in all seasons (Portmann et al., 2021). The wind speed anomaly at 300 hPa shows a well-defined cyclonic circulation with a pronounced easterly flow anomaly over Ukraine, in agreement with the equatorward gradient of the PV anomaly in this region.

Last, for EPEs in autumn (Fig. 5d), the positive PV anomaly is strongly meridionally oriented similarly to spring, but now extends directly over Ukraine. Negative PV anomalies are found over southern Poland and Slovakia, creating an eastward gradient of PV anomalies over western Ukraine and an anomalous northerly flow, leading to the emergence of orographically enhanced EPEs in Transcarpathia. Over the Black Sea and Eastern Ukraine, there is an enhanced southerly flow and the pronounced positive PV anomaly most likely contributed to the intensification of cyclones over eastern Ukraine (see Fig. 3d). The standardized anomaly pattern exhibits a seasonal variation, reaching its peak (approximately 1.7 SD) during winter and reaching a  minimum of 0.7 SD in summer, in the main PV anomaly regions (supplementary Fig. S2). We note, however, that these fields should be regarded with caution in all seasons except summer, because of the low number of events.

In summary, during all seasons EPEs in Ukraine are associated with pronounced upper-level PV anomalies. As a common feature, in all seasons, the region of Ukraine is located between positive and negative PV anomalies. However, interestingly, the orientation of these anomaly dipoles differs strongly between the seasons, and can be classified, to first order, as northward in winter, westward in spring, southward in summer, and eastward in autumn. Consistent with the basic understanding of PV dynamics, these differently orientated PV anomaly dipoles lead to characteristics seasonal patterns of the anomalous upper-level flow, and also can influence the moisture transport process in the middle troposphere. In each season, EPEs appear to be preconditioned largely by a moist flow from the southwest, south, or southeast, along the eastern flank of the upper-level PV anomalies.

**3.4 Seasonal mean moisture sources**

To categorize and summarize the various moisture source contributions of EPEs in Ukraine, we define large-scale source regions, separately for oceanic and terrestrial sources. As oceanic moisture sources, we include the midlatitude North Atlantic, the Mediterranean Sea (western and eastern parts, separately), the Black and Azov Seas, and the Caspian Sea. Terrestrial regions considered are western and eastern Europe, Italy and the Balkans, Ukraine, the East European Plain, Africa, and Asia (Fig. 6a). Figures 6b and 6c provide information about the percentage contribution from different moisture

sources for EPE in all seasons, and seasonal moisture uptake composites are shown in Fig. 7.

In winter, EPEs in Ukraine have predominantly oceanic moisture origins (67%, Fig. 6b). An elongated uptake zone is located

over the midlatitude North Atlantic (24%), in the western (18%) and eastern Mediterranean (12%), and the Black Sea (7%),

consistently with the strongly enhanced westerly flow discussed in Sect. 3.2. The share of terrestrial sources (34%) is smaller

than the oceanic contributions. The main land sources are western Europe (8%), the Balkans (6%), and Asia (6%). The

maximum moisture sources are located over the North Atlantic, the Mediterranean and the Black Sea.

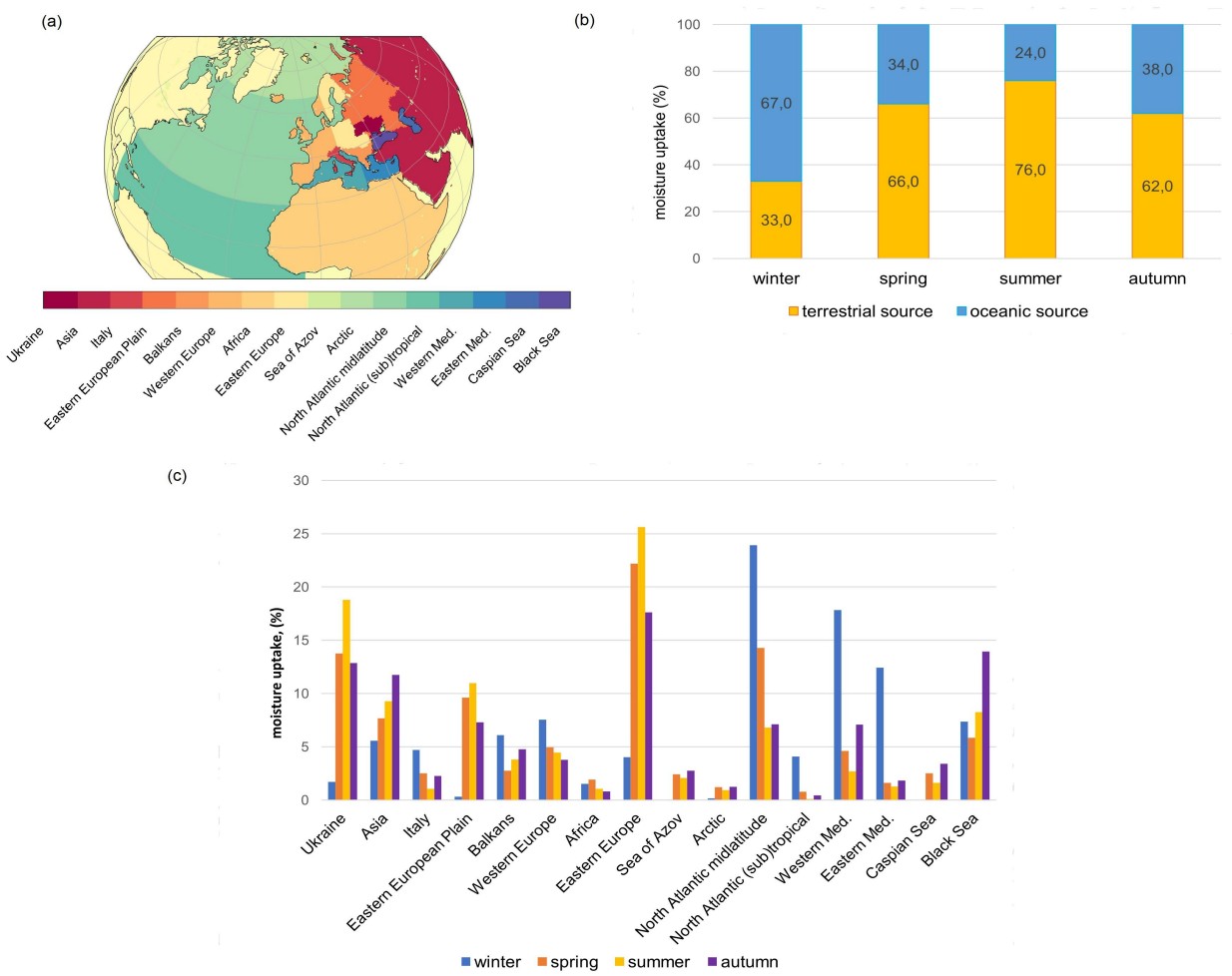

**Figure 6 (a) Predefined moisture source regions; (c) shows their seasonal-mean relative contributions on EPE days**
**in Ukraine, and (b) shows these contributions aggregated to terrestrial and oceanic sources.**

In the other seasons, the moisture sources are predominantly over land (Fig. 6b). In spring, the total moisture contribution

from land surfaces increased to 66% (Fig. 6b), with local contributions of 22% from eastern Europe, and 14% of continental

recycling over Ukraine. The maximum moisture source is located over southern Ukraine and the Azov Sea (Fig. 7). A

substantial eastern footprint also emerges from the East European Plain and Asia with 17%. The oceanic contributions from

the North Atlantic are 14% (compared to 24% in winter), and evaporation from the Black Sea provides 6% (similarly as in

winter). In the east, the Caspian Sea becomes a relevant moisture source with 3%. Some remote sources are also identified

over western Europe, Italy and the Balkans, but they are much weaker than those over eastern Europe and Ukraine.

In summer, contributions from remote moisture sources to EPEs in Ukraine are strongly reduced and evapotranspiration

from land is clearly the dominant source with 76% (Fig. 6b). Main local contributions are from eastern Europe (26%),

Ukraine (19%), and the Eastern European Plain (11%). The 24% of oceanic moisture sources of summer EPEs were diagnosed from the Black Sea (8%), the midlatitude North Atlantic (7%), Western and Eastern Mediterranean (4%). Moisture uptake from the Caspian Sea was weaker than in spring and autumn (2%). EPEs in autumn have also mainly continental moisture sources      (62 %), mainly from eastern Europe (18%), Ukraine (13%), and Asia (12%). The influence of oceanic moisture sources from western Mediterranean increases slightly compared to summer. The Black Sea becomes a very important moisture source in this season with a 14% contribution. Also, considerable continental moisture recycling is identified in the target region of the EPE, i.e. in the southern Ukraine. The maximum uptake is located around Crimea. And finally, moisture uptake from the Caspian Sea was the largest compared to the other seasons (3 %), most likely consistent with advection from the east associated with the strongly negative MSLP anomalies in eastern Ukraine (Fig. 3d).

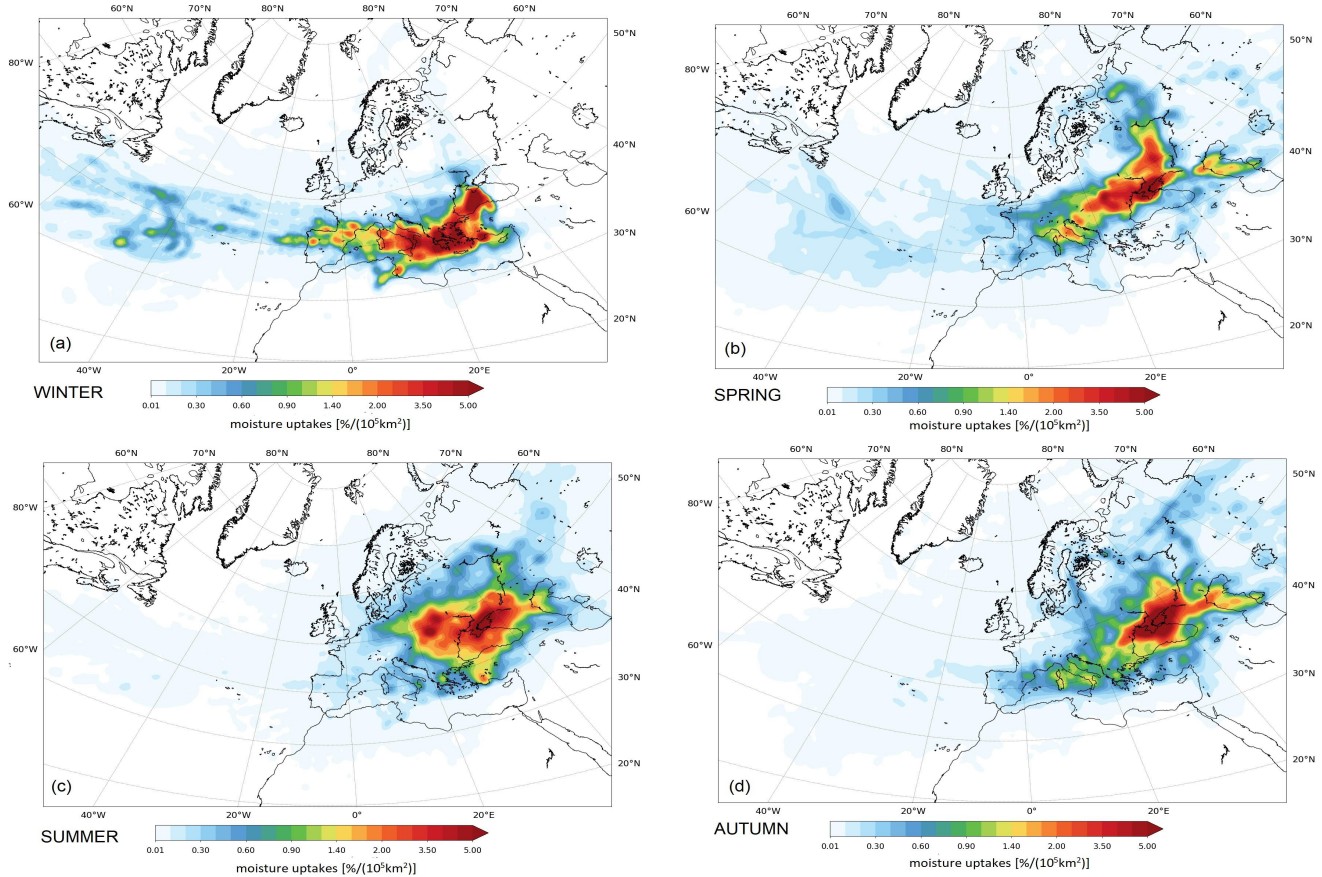

**Figure 7 Seasonal mean moisture sources for EPEs in Ukraine (%/$10^5$ km$^2$).**

It is noteworthy, that there is less coherent structure in the fields of moisture sources compared to the upper-level circulation fields investigated in the previous sections. This may be due to the fact that the upper-level circulation is often governed by large-scale flow features, for example, the presence of a strong jet stream or a well-defined upper-level trough. This can explain their somewhat more consistent structure compared to the more variable moisture sources. Since by far most of the global water vapor is located in the lower troposphere, moisture source fields are influenced by factors like sea surface temperatures, local evaporation, soil moisture availability, moisture transport, and low-level winds, and convection. Winschall et al., (2014) investigated the importance of intensified local and remote evaporation for Mediterranean precipitation extremes. Krug et al. (2022) determined that the evaporation anomalies are related to wind-speed anomalies indicating mainly dynamically driven evaporation. Grams et al. (2014) emphasized the significant role of soil moisture

preconditioning. For instance, intense precipitation events can moisten the previously dry soil and might subsequently serve

as moisture sources for subsequent extreme EPEs (Bohlinger et al., 2017). And lastly, Dahinden et al. (2023) studied shallow

and deep convective systems that occur in random patches and lead to highly variable structure to the moisture source maps.

This complex interaction between various preconditioning factors and the eventually emerging moisture source patterns

should be investigated in more detail in future research.

In summary, this overview on seasonal moisture sources that contribute to EPEs in Ukraine reveals a large variability of the

sources, including local recycling and long-range transport over several 1000 km for instance from the central North Atlantic

(in winter and spring) and from the Caspian Sea (from spring to autumn). Oceanic moisture sources dominate in winter and

land moisture sources in all other seasons. Given that most EPEs in Ukraine occur in summer (Sect. 3.1) it becomes clear

that local recycling over Ukraine and land evapotranspiration over the neighboring regions (eastern Europe and the East

European Plain) are very important for understanding EPEs in Ukraine. And, in summer, the contributions from the Black

Sea are greater than those from the Mediterranean. This conclusion reflects that moisture fields can display high variability

and are influenced by a range of dynamic and local factors.

The principal results of the analysis of seasonal EPE characteristics in Ukraine are summarized in Table 1 to enhance clarity

and facilitate comparison. This allows us to contrast the results across seasons.

**Table 1 Mean seasonal characteristics of EPEs in Ukraine: Anomalies (units), moisture sources (MS, %) and affected regions. Table abbreviations: W-west, E-east, S-south, N-north.**

| | | Winter | Spring | Summer | Autumn |
|---|---|---|---|---|---|
| $Z_{500}$ | | Upper-level trough over E Europe & Ukraine:-241 m | Upper-level low over E & S Europe, Ukraine: -48 m | Upper-level low over the Balkans, the Black Sea & Ukraine: -43 m | Upper-level low over the Black Sea & Ukraine: -79 m |
| MSLP | | Low over the Balkans: -24 hPa SE Ukraine: -20 hPa | Low over W & SW Ukraine: - 6 hPa | Low over Ukraine: - 4 hPa | Low over the Black Sea & E Ukraine: - 12 hPa |
| PV (+) | | Isentropic levels | | | |
| | | 315 K | 325 K | 335 K | 330 K |
| | | PV(+) north of $40^0$N: 0.1-2.5 PVU Cutoff over NE Black Sea region: 0.2-0.5 PVU | PV(+) over E Europe & W Ukraine:1-2 PVU | Cutoff over the Balkans, Black Sea & SW Ukraine: 0.2-1 PVU | PV(+) over the Baltic region, N, Central & SW of Ukraine: 0.2-1 PVU; Cutoff over E Ukraine: 0.2-1 PVU |
| $U_{300}$ | | SW jet stream over Mediterranean: +35 m·s$^{-1}$, the Balkans: +30 m·s$^{-1}$ Ukraine, the Black & Azov Seas: +15-25 m·s$^{-1}$ | N jet stream over E Europe, the Balkans & W Ukraine: +10 m·s$^{-1}$ | Negative anomaly over W Ukraine: −10 m·s$^{-1}$ Positive anomaly over the Balkans & the Black Sea: +5 m·s$^{-1}$ | SW jet stream over the Black Sea & E Ukraine: +10 m·s$^{-1}$ |
| MS | | 1. North Atlantic (24 %), 2. Western Med (18 %) 3. Eastern Med (12 %) 4. Black Sea (7 %) | 1. Eastern Europe(22 %) 2. North Atlantic (14 %) 3. Ukraine (14 %) 4. East European Plain & Asia (17 %) 5. Caspian Sea (3%) | 1. Eastern Europe (25 %) 2. Ukraine (19 %) 3. East European Plain (11 %) 4. Black Sea (8 %) 5. North Atlantic (7 %) 6. Caspian Sea (2 %) | 1. Eastern Europe (18 %) 2. Ukraine (13 %) 3. Asia (12 %) 4. N Atlantic & W Med (14 %) 5. Black Sea (14 %) 6. Azov Sea (3 %) 7. Caspian Sea (4 %) |
| EPE regions | | W Ukraine, Crimea | W Ukraine, Crimea | Entire territory of Ukraine | W, SW & SE Ukraine, Crimea |

**3.5 Case studies of selected EPEs**

After the climatological overview on EPEs in Ukraine given in the previous subsections, it is important to also show representative case studies of EPEs to obtain a more detailed understanding of the dynamics and associated moisture sources leading to the occurrence of these meteorological hazards. To this end, we selected eight events, two in each season. They are: (i) 28 December 1999 on the Crimean Peninsula; (ii) 21 December 1993 in western Ukraine; (iii) 15 May 2014 in the Transcarpathian region; (iv) 31 May 2014 in eastern Ukraine; (v) 1 July 2011 in central Ukraine; (vi) 1 August 2019 in southeastern Ukraine; (vii) 24 September 2014 in the Crimea; and (viii) 12 October 2016 in the northwestern Black Sea region. For each case, we briefly discuss the patterns of MSLP and surface precipitation, geopotential height at 500 hPa, and the identified moisture sources. Table S2 in the Supplement lists the relative contributions of the different moisture sources (Fig. 6a) for these cases.

**3.5.1 Winter cases:  28 December 1999 and 21 December 1993**

Both winter cases occurred under strong westerly flow and show common features and some distinctively different characteristics. The first EPE, on 27 and 28 December 1999, occurred two days after the infamous winter storm 'Lothar' (24–26 December 1999) strongly damaged parts of France, Germany, and Switzerland. This storm developed beneath an exceptionally intense and zonally elongated westerly jet over the North Atlantic with wind speeds up to 120 m·s$^{-1}$ (Wernli et al., 2002). In the next days, a series of cyclones moved from southeastern Europe over the northern coast of the Black and Azov Seas. One of the cyclones of this series caused extreme precipitation in the Crimean Peninsula. On  27 December, 112.5 mm·day$^{-1}$ were observed at Ai-Petri, and on 28 December 100 mm ·day$^{-1}$ at Yalta with minor wave disturbances and 115.3 mm·day$^{-1}$ at Ai-Petri. The cyclone formed on 27 December in a short-wave perturbation over the Lower Danube Plain and the Black Sea Lowland between a deep Scandinavian low-pressure system and a high-pressure zone to the south. During 24 h it intensified rapidly and attained its minimum pressure of  990 hPa over Crimea and the Azov Sea. The largest precipitation values were registered close to the center of the cyclone (Fig. 8a). At upper levels there was an intense zonal flow with minor wave disturbances (Fig. 8b), which can be regarded as the extension of exceptional North Atlantic jet that led to the development of 'Lothar'. It is remarkable, that one of the rare and most intense winter EPEs in Ukraine occurred right after one of the most severe winter storms in western and central Europe. The moisture sources for this EPE were mainly around Greece but extend in a zonal band far upstream into the central North Atlantic (Fig. 8c), i.e., in the region of rapid propagation of 'Lothar'. Notable contributions were from the North Atlantic midlatitude (16-28 %), Western Mediterranean (19-24 %), Eastern Mediterranean (11-14 %) and the Black Sea (7-14 %).

The second winter EPE was on 21 December 1993 (Fig. 8d). During this event precipitation concentrated over the Transcarpathian region. The station Rahiv recorded 101.4 mm day$^{-1}$. The EPE was influenced by a surface cyclone that formed over southern Poland. As for the first case, an intense baroclinic zone with a strong upper-level zonal flow extended from the eastern North Atlantic in this case to the Caspian Sea. This went along with a deep low-pressure system over Scandinavia, the Baltic regions, the Kara Sea, and a high-pressure system over southern Europe and the Mediterranean (Fig. 8 d,e). The moisture sources were again extended far into the North Atlantic, in this case also with a substantial contribution from the subtropics. Other moisture sources were over continental areas of Europe and Ukraine (Fig. 8f). The largest contributions were from the North Atlantic mid- and subtropical latitudes (39% and 18%, respectively), and Western Mediterranean (14%). The terrestrial moisture sources, specifically Eastern Europe and Western Europe, made relatively minor contributions, accounting for 13% and 9%, respectively. Overall, long-range advection of oceanic moisture

contributed a major part to the winter EPEs. The percentage of oceanic moisture contributions was 54% on 28 December 1999 and 68% on 21 December 1993.

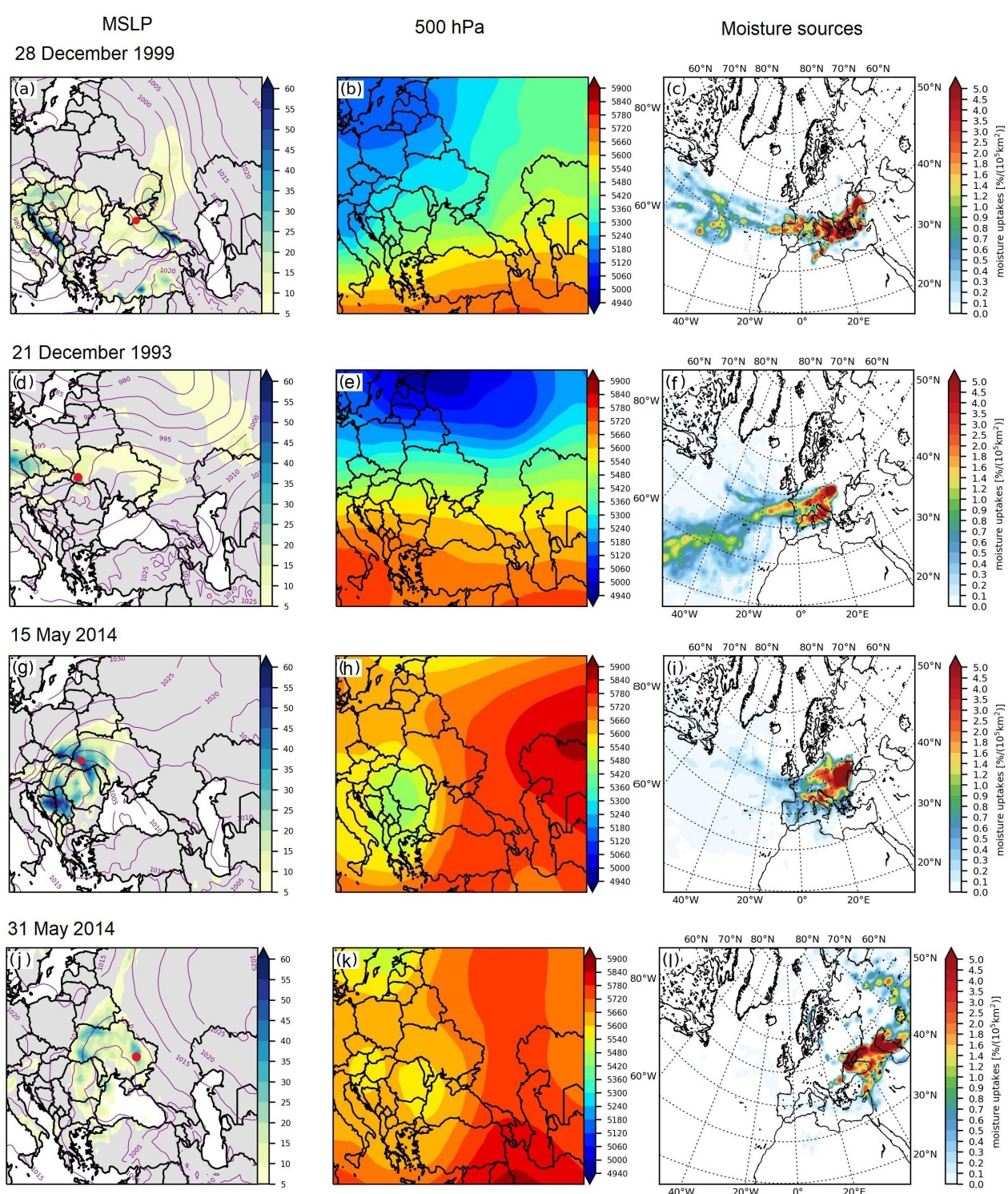

**Figure 8** Overview on four EPE case studies (see dates on top of left panels), based on ERA-5. (a,d,g,j) show MSLP at 2100 UTC (purple contours, every 5 hPa) and daily accumulated total precipitation (mm, color shading), the red dot indicates a station with precipitation > 100 mm·day$^{-1}$; (b,e,h,k) show 500-hPa geopotential height at 2100 UTC (color shading, in m); and (c,f,i,l) show moisture uptake regions (in %/($10^5$km$^2$).

**3.5.2 Spring cases:  15 May 2014 and 31 May 2014**

Both considered spring EPEs occurred in May 2014. They are interesting in that they affected different parts of Ukraine and had different moisture sources, despite a quite similar mid-tropospheric configuration. Between 14-16 May 2014, an EPE occurred in the Carpathians and Transcarpathia, with recorded values ranging from 106-145 mm·day$^{-1}$, which corresponds to more than the monthly average. Concurrently, strong wind gusts exceeding 19 m·s$^{-1}$ were observed. These weather conditions led to severe flooding in the Dniester River basin. Additionally, the heavy rainfall triggered mudslides, affecting a total of 94 settlements, as documented by the European Severe Weather Database (ESWD, Dotzek, 2009). On 15 May, extended precipitation was observed in the Transcarpathian region, and 104.7 mm·day$^{-1}$ were recorded at Yaremche. The precipitation was caused by a deep cyclone (with a MSLP minimum of 1000 hPa) that formed over the Balkans on 14 May and reached Western Ukraine on 15 May (Fig. 8g).  During the mature stage, an upper-level trough with a deep core formed over southeastern Europe and on 15 May overlapped with the surface cyclone (Fig. 8h). A large upper-level anticyclone in the east shaped a high-pressure belt over most parts of Ukraine and had a blocking effect, inhibiting further shifting of the cyclone to the northeast, which caused widespread precipitation over western Ukraine. Figure 8i shows that moisture sources for this EPE were mainly over Eastern Europe, Ukraine, Western Europe, and the Balkans with contributions of 37%, 19%, 8%, and 8%, respectively. The North Atlantic plume of moisture contributed with 11%.

The second spring EPE occurred two weeks later, but in the east of Ukraine. On 31 May, 104.4 mm ·day$^{-1}$ were recorded at Lozova. Damage was reported due to flooding, also to crops (ESWD, Dotzek, 2009). Precipitation was observed in a wide frontal band that formed between a cyclone that developed over Ukraine and the Black Sea with simultaneous intense anticyclogenesis over the East European Plain (Fig. 8j). At upper levels, a stationary trough extended from the north over East Europe and Ukraine with two centers of low pressure with similar intensity over Austria and Hungary, and over Bulgaria and Moldova (Fig. 8k). A strong ridge extended northward from Minor Asia and the Caspian Sea and again had a blocking-like signature affecting east Ukraine (Fig. 8k). In strong contrast to the previous three EPEs, moisture sources for this case were interestingly mainly further east. They stretched from the West Siberian Plain to the east Ukraine, the Black Sea, and east Turkey. The largest contributions were from Asia (28%) and Ukraine (15%). The moisture uptakes over the Caspian Sea, the Black Sea and the Azov Sea accounted for 15%, 13% and 6%, respectively (Fig. 8l).

**3.5.3 Summer cases:  1 July 2011 and 4 August 2019**

The first selected EPE occurred on 1 July 2011, with precipitation spreading across the north of Ukraine. At Barishevka (Kyiv region) 130.5 mm·day$^{-1}$ were observed. Damage occurred due to flooding of local areas (ESWD, Dotzek, 2009). Two weak surface cyclones developed below an upper-level trough extending from Northern Europe, one over northeastern Ukraine and the other one east of the Black Sea, with central MSLP values of 1005 and 1010 hPa, respectively (Fig. 9a). An EPE formed in the northern regions of Ukraine along a cold front. Again, a blocking effect was exerted by a large anticyclone over the East European Plain. At upper levels, a stationary ridge associated with that surface high-pressure system spanned from the Middle East and Central Asia toward the north (Fig. 9b). Long-range transport of moisture is evident from three bands of moisture sources (Fig. 9c) A substantial amount of terrestrial moisture originated over Asia (32%) and the East European Plain (29%). Two other, much weaker, branches were formed over the Caspian Sea (2%) and the Black Sea basin (4%). Moisture uptake over Ukraine contributed with 19%.

The second summer EPE on 3-4 August 2019 was associated with heavy precipitation propagating across the southwest to the northeast of Ukraine along strong frontal systems associated with a cyclone moving from the Balkans towards eastern

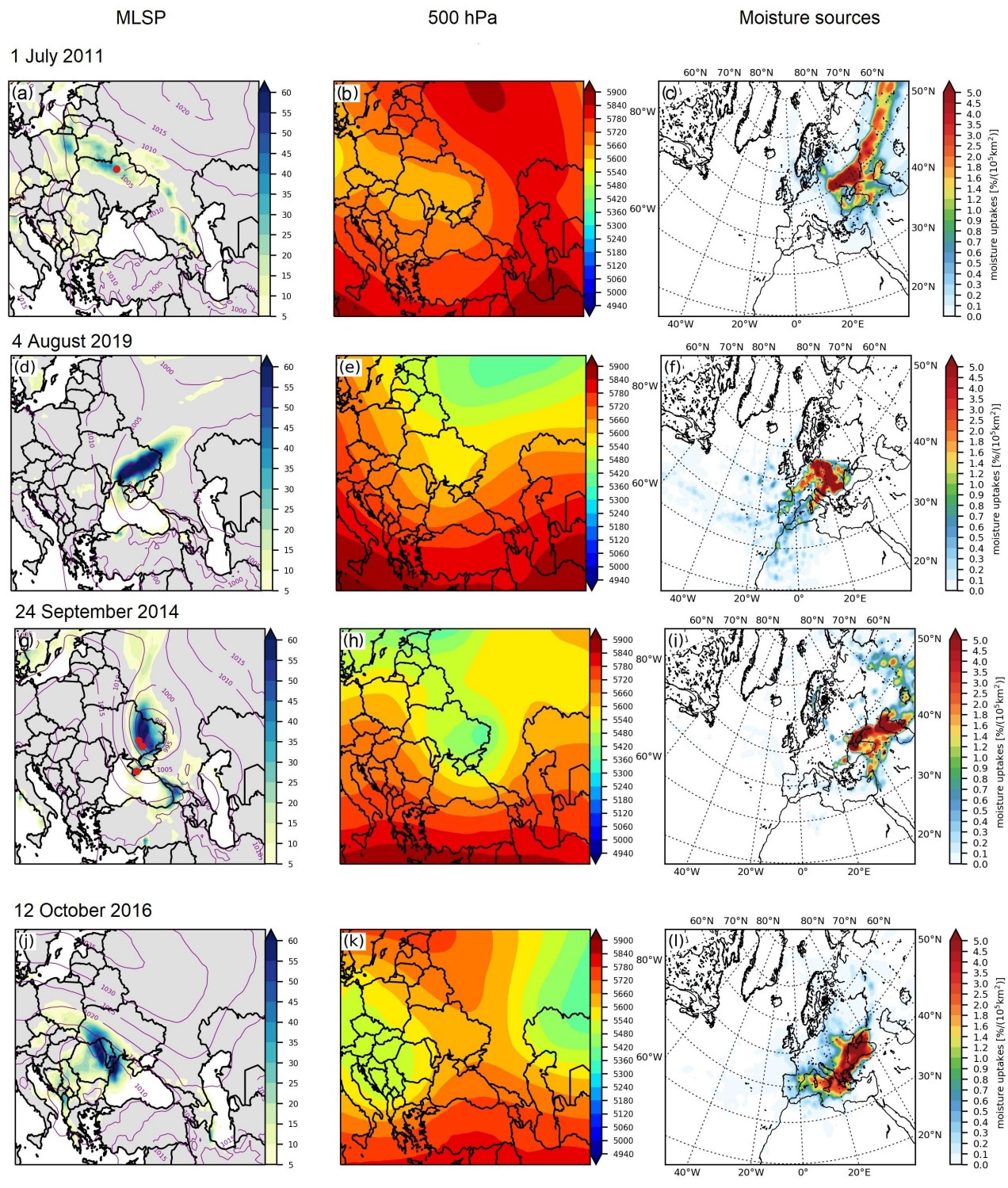

| MLSP | 500 hPa | Moisture sources |
| --- | --- | --- |

1 July 2011

4 August 2019

24 September 2014

12 October 2016

**Figure 9 The same as Fig. 8 but for two EPEs each in summer and autumn (dates are indicated again on top of the**
**left panels).**

Ukraine (Fig. 9d). Combined with strong winds (15-24 m s$^{-1}$), this caused urban flooding and damaged energy infrastructure

in the regions of Odesa, Kherson, Zaporozhye, Donetsk, Dnepropetrovsk and Lugansk (ESWD, Dotzek, 2009). Extreme

precipitation was recorded at Belgorod Dnestrovsky in the Odesa region and at Khorli in the Kherson region with 125.2

497 mm·day$^{-1}$ and 106.4 mm·day$^{-1}$, respectively. In this case, a pronounced upper-level trough extended from northern Russia

through Ukraine towards the Black Sea (Fig. 9e). A wide baroclinic zone occurred along the southern edge of the trough, stretching from southeastern Europe over the Black Sea to Middle Asia. This caused the formation of a strong northwesterly flow that advected relatively cool air to the Balkans, Turkey, and the western Black Sea. At the same time, warm air of tropical origin from Minor Asia and Caucasus propagated across the southeast of Ukraine and Crimea. On 3 August, the surface cyclone formed in a short-wave perturbation over the Balkans. The cyclone rapidly intensified and moved eastwards, made landfall in western Crimea 24 h later, where it reached its minimum MSLP below 995 hPa, and further passed on the eastern Ukraine. This EPE was characterized by a predominance of land evapotranspiration, accounting for 83% of the moisture. Notably, strong moisture contributions were observed in a large area of East Europe (48%) with additional moisture from West Europe, the Balkans and recycling over Ukraine. In contrast, oceanic contributions were relatively minor, with 9% from the North Atlantic midlatitude and 6% from the western Mediterranean (Fig. 9f).

**3.5.4 Autumn cases: 24 September 2014 and 12 October 2016**

On 23-24 September 2014, precipitation was observed mainly over southeastern and eastern Ukraine (Fig. 9g). Extreme precipitation was recorded at three stations: Prishib (Zaporizhzhia region, 114.7 mm·day$^{-1}$), Sinelnikove (Dnipropetrovsk region, 100.1 mm·day$^{-1}$), and Ai-Petri (Crimea, 107.8 mm·day$^{-1}$). Major damage was caused by the strong winds (25 m s$^{-1}$) and heavy precipitation (ESWD, Dotzek, 2009). A trough from north Russia towards the Black Sea developed on 22 September and a deep closed cyclone over Crimea and the Azov Sea formed there on 23 September (Fig. 9h). This cutoff low system then propagated over Ukraine and the associated surface cyclone intensified strongly with central MSLP decreasing to 985 hPa (Fig. 9g) – the most intense cyclone in the considered case studies. Two stationary anticyclones, located over central Europe and over Russia, most likely exerted a blocking effect. The EPE resulted from a complex set of moisture sources (Fig. 9i). The main moisture sources were found over Asia (27%), the Eastern European Plain (18%), the Caspian Sea (18%), Ukraine (15%), and the Black Sea (13%).

The second autumn EPE occurred on 12-13 October 2016 with strong winds exceeding 25-31 m s$^{-1}$ and heavy precipitation over the Odesa region with 103 mm·day$^{-1}$ at Bolgrad on 12 October. Damage and four fatalities were reported due to winds and urban flooding (ESWD, Dotzek, 2009) and a state of emergency was declared in Odesa on 12 October. During this period, a quasi-stationary intense anticyclone was located over Scandinavia extending toward the Caspian Sea through most of the European part of Russia (Fig. 9j). At the same time, a cyclone developed over southern Europe, intensified to a MSLP minimum of 1005 hPa, and moved towards the northwest of the Black Sea. The precipitation area associated with a strong frontal system was extended along southwestern Ukraine and Moldova (Fig. 9j). On 12 October, a narrow upper-level ridge elongated over most of Ukraine, flanked by two upper-level cyclones (Fig. 9k). An intense baroclinic zone formed over southwestern Ukraine, within which the cyclone resided over the Odesa region for two days (not shown). This EPE shows a continuous band of moisture sources from the western Mediterranean to the Black Sea and south Ukraine (Fig. 9l). This event had a relatively large Black Sea moisture contribution (22%). The Black Sea is still quite warm in autumn, increasing the potential for intense evaporation. Other moisture sources for this event were mainly the Western (16%) and Eastern Mediterranean (9%), and the Balkans (12%).

Thus, this analysis of the large-scale flow conditions and moisture sources for eight different EPEs reveals a large variability from case to case. However, it is important to highlight that all EPEs, except those in winter, were influenced by a pronounced upper-level trough over Ukraine and a high-pressure system east or north of Ukraine. The most intense precipitation occurred during the EPE on 24 September 2014, when a cutoff formed and remained stationary over the target area. In stark contrast, the winter EPEs occurred in situations with exceptionally strong westerly jets. The local trough

configuration predominantly facilitated moisture sources of terrestrial origin and led to precipitation recycling over Ukraine during the EPEs days, whereas the winter EPEs had important long-range transport from the (subtropical) North Atlantic.

**4 Summary and conclusions**

This study presents results of a climatological investigation of EPEs in Ukraine in the period 1979-2019. EPEs were identified with precipitation exceeding a simple threshold of 100 mm·day$^{-1}$ at measurement stations, and ERA5 reanalyses were used to investigate the large-scale physical and dynamical processes that were involved in the formation of these EPEs. In the following, provides a summary of the main results and the basis for addressing the four main aspects of EPEs investigated in this study, which are (1) the seasonal occurrence, frequency, and spatial distribution of EPEs in Ukraine, (2) the dynamical characteristics during EPEs, (3) the origin and transport pathways of moisture that led to the EPEs, and (4) the variability between individual cases.

1. Results show that Ukraine has two hotspots of EPE frequency: the Ukrainian Carpathians and Crimea. EPEs were recorded in all seasons in those regions. Nevertheless, in summer, during the season of maximum frequency of EPEs, they were observed not only in mountainous regions, but also across most other parts of Ukraine. In autumn, EPEs prevailed on the northwestern and northeastern coasts of the Black Sea.

2. EPEs occur due to relatively rare and anomalous circulation processes. Analysis of a combination of SLP and Z500 anomalies, upper-level PV and 300-hPa winds has shown that: (i) negative anomalies of SLP and Z500 were found in all seasons, and PV streamers and cutoffs on 315–330 K occur in the key areas of cyclogenesis over Ukraine; (ii) anomalies of SLP and Z500, PV and 300-hPa wind show a clear connection with the observed EPEs over most of the studied domain, and with anomalies in total column water and, only in summer, in CAPE; (iii) isentropic potential vorticity anomalies associated with EPEs in Ukraine show distinct dipole patterns which changes from one season to the other, rotated by 90 degrees: northward in winter, westward in spring southward in summer, and eastward in autumn; (iv) winter, spring and autumn anomalies were distinguished by higher intensities compared to summer, however, EPEs were most frequently registered in summer and over the entire Ukraine. This might imply that, during summertime, the occurrence of EPEs in Ukraine is modulated not only by the large-scale circulation, but also by localized convection, which can play a significant role in shaping EPEs during this period.

3. The moisture source regions for the EPEs in each season in Ukraine have been investigated with a trajectory-based Lagrangian moisture source diagnostic. The results show that EPEs mainly in winter were associated with long-range atmospheric moisture transport of oceanic origin, which occurred southward of the maximum positive PV anomaly region. Moisture uptake regions were the subtropical and midlatitude North Atlantic, and the Mediterranean. However, during the other seasons, terrestrial moisture sources dominated in contributing to EPEs. In spring and autumn, the moisture contributions from land surfaces represented mainly a combination of different local sources and additional remote sources, both from the European continent and from Asia. Evaporation from the North Atlantic in spring and from the Mediterranean Sea in autumn, in combination with transport from the Caspian Sea provided moisture from ocean sources during those seasons. A correlation with PV dipoles localization was also observed: the predominance of moisture flows from remote sources on the southeastern flank of the positive PV anomaly in spring and along the southwestern edge in autumn. In summer, the primary source of moisture over Ukraine was land evapotranspiration, mainly from Eastern Europe, Ukraine, and the East European Plain, and the area of maximum moisture uptake practically overlapped with the region of positive PV. It is worth noting the contribution of the Black Sea as a local

source of moisture, which is an important oceanic source region that provided year-round moisture to EPEs along the
south coast, as well as some continental regions of Ukraine.

4.   Analysis of large-scale flow conditions and moisture source regions for individual events, based on ERA5 data, has
confirmed that EPE generation in spring, summer, and autumn was mainly due to the impact of upper-level troughs
extending over eastern and southern Europe and blocking anticyclone over the Eastern European Plain. In the western
and southwestern regions of Ukraine, cutoffs formed during some EPEs. The exception were winter EPEs, when the
cyclones formed due to a short-wave perturbation in the westerly flow, which delivered moist air from the North
Atlantic and the Mediterranean to western and southern Ukraine. The study of the moisture sources for eight EPEs in
Ukraine showed important case-to-case variability. This indicates, very importantly, that seasonal mean conditions are
not necessarily representative for individual EPEs, and that even two EPEs occurring in the same month (see the two
EPEs selected in May 2014) can have very different moisture sources despite relatively similar patterns in 500-hPa
geopotential height. Clearly this field, often investigated in synoptic climatologies, cannot fully represent the complex
dynamics and moisture transport at multiple scales involved in EPEs.

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

**Data availability**
ERA5 data is openly available at https://cds.climate.copernicus.eu (Hersbach et al., 2020). The observational data used for this study can be requested from the Central Geophysical Observatory in Ukraine (http://cgo-sreznevskyi.kyiv.ua/en/ ).

**Author contributions**
EA and HW designed and planned the study, EA performed the analysis, and wrote the manuscript with support from HW. Visualizations were produced by MA, FA, AS and EA. All authors contributed to the interpretation and discussion of the results.

**Competing interests**
The authors declare that they have no conflict of interest.

**Acknowledgements**
EA is grateful to OSENU (Odesa State Environmental University) for providing the precipitation observations from Ukraine. The authors also thank the reviewers for their constructive feedback that helped improving the presentation of the results.

**Financial support**
EA and AS were supported by the Swiss National Science Foundation (grant no. 212026 and 216775, and 188660, respectively). MA was supported by an ETH Zürich postdoctoral fellowship (project no. 21-1 FEL67) and by the Stiftung für naturwissenschaftliche und technische Forschung as well as the ETH Zürich Foundation.