# Peer review of "Precipitation extremes in Ukraine from 1979 to 2019: Climatology, large-scale flow conditions, and moisture sources"

_EGUsphere, 2023_

## Author Comment (AC1)

*Paper egusphere-2023-2594*

**Precipitation extremes in Ukraine from 1979 to 2019: Climatology, large-scale flow conditions, and moisture sources**

by Ellina Agayar, Franziska Aemisegger, Moshe Armon, Alexander Scherrmann, and Heini Wernli

*Final author comments*

We thank all three reviewers for their thoughtful and constructive comments that help us to improve the manuscript. Based on the reviewers' suggestions, we implement several changes in the manuscript. The main changes are that:

- We add a new figure to show more insight into the EPEs in summer (in response to suggestion by reviewer #3).
- We further clarify the data availability (in response to comment by reviewer #3).
- We add more discussion about the moisture source composites in comparison to those for upper-level PV (in response to comment by reviewer #2).
- We better motivate the choice of the 100 mm per day threshold to identify EPEs (in response to comment by reviewers #1 and 2).

Below we provide a one-to-one response to all points raised by the reviewers. The reviewers' comments are in red and our replies in black.

**Reviewer #3**

Overall, I found this manuscript to be clear, concise, and well-written. The study provides a novel climatological investigation of extreme precipitation events (EPEs) in Ukraine, documenting the synoptic-scale conditions in which these events occur and quantifying the moisture sources using a Lagrangian trajectory-based diagnostic. The findings help to address a gap in scientific understanding regarding EPEs in Ukraine.

While the paper is strong overall, I have a number of comments for the authors to consider. Once these comments are satisfactorily addressed, the manuscript may be acceptable for publication.

We express our sincere gratitude to Reviewer #3 for their comprehensive analysis of our work and for providing valuable comments and suggestions for our paper. We are confident that integrating solutions to these specific questions into the revised version of our manuscript will significantly improve its overall quality.

**Major comments**

In my opinion, the manuscript is lacking in diagnostic analysis of the ingredients and processes resulting in extreme precipitation. Composite analyses and case studies are

presented, and the circulation patterns are discussed, but it is still not entirely clear to me how the ingredients for heavy precipitation are established and maintained for these events. Are these events characterized by, for example, particularly anomalous moisture content or strong ascent? Do the key flow features tend to be particularly slow-moving? The study could be strengthened in this regard by inclusion of additional statistical/composite/case study analyses of key ingredients, such as dynamical forcing for ascent (e.g., quasi-geostrophic forcing or frontogenesis), moisture content/moisture flux, and convective instability (e.g., convective available potential energy). Such analyses would help to elucidate how the circulation features shown in Figs. 3 and 4 are linked to the ingredients for heavy precipitation in Ukraine, thereby providing a more complete picture of the synoptic-scale characteristics of the EPEs. Precipitation composites based on the ERA5 could also help to show where in the region the precipitation tends to be focused for the different seasonal groups of EPEs, thereby providing helpful context when interpreting the composite patterns.

We agree with the reviewer that a more in-depth analysis is useful, and we added analyses of composites of total precipitation, TCW and CAPE for a more complete picture of the synoptic-scale characteristics of the EPEs in Ukraine to the Sect. 3.2. The figure below (Fig. R1; similar to Fig. 4) shows anomalies of the total precipitation, TCW and CAPE in physical units at 15 UTC for EPE days in summer. (However, to avoid overloading the paper with additional figures, we added similar figures to the Supplement for the other seasons). The figure shows anomalously high values of TCW and CAPE over the entire Ukraine on summer EPE days.

[Figure]

**Fig. R1 Anomalies of total precipitation, TCW and CAPE at 15 UTC on EPE days in summer.**

How was the 100 mm day$^{-1}$ threshold selected, and how extreme is it for the various stations? I recommend quantifying where this threshold fits in the climatological distribution at each station. Would it be possible to identify EPEs as daily precipitation totals exceeding an upper percentile (e.g., 95th percentile) of the climatological distribution for each station instead of a fixed threshold?

See our response to minor comment 1 of reviewer #2. Note that with our choice of the threshold we identify less than 100 EPEs at about 200 stations over 40 years. Therefore, the corresponding percentile is much larger than the 95$^{th}$ percentile mentioned by the reviewer, indicating that this study focuses on extreme, i.e., very rare events.

There is redundancy in showing maps of both 500-hPa geopotential height and near-tropopause PV. Both fields depict qualitatively similar structures and patterns of the upper-level flow. Is it necessary to show both fields?

We agree that there is some redundancy between these fields, but as discussed in many studies about the usefulness of upper-level PV charts (e.g., Hoskins et al., 1985), the PV charts show more structural details that can be useful to understand the dynamics. So far, studies showing PV charts have not been conducted for the Ukrainian domain. On the other hand, the choice of the classical 500-hPa charts is motivated by the fact that it has been widely use in synoptic analyses. Thus, to show the similarities and differences of the two fields, we think that both fields are quite useful for our paper.

**Minor comments**

Line 82: Do all of the stations have the same record length? Are they all available for 1979–2019?

Thanks for this question, it's really worth covering this point in more detail.

For this study, 215 meteorological stations and posts (including aviation weather stations, gauging stations, etc.) with daily data from 1979 to 2019 are used. From this dataset, 183 stations were selected for our study due to having a complete set of data. The remaining 32 stations did not have the same record length for various reasons. Nevertheless, these stations were still tested for the presence of extreme precipitation events (EPE) $\geq$ 100 mm day$^{-1}$, and no such extreme events were recorded. Due to the absence of data in the Ukrainian meteorological network for certain regions of Crimea from February 2015 to December 2019, additional data were obtained using open-access observations for this region (SYNOP observational data). Unfortunately, data for four stations in the Donetsk and Lugansk regions for the period of 2015-2019 are not openly available. In this region, a 36-year dataset was employed to identify days with extreme precipitation ($\geq$ 100 mm day$^{-1}$). We will add this more detailed explanation of the utilized dataset in the revised manuscript.

Line 88: It would be more accurate to state that the reanalysis data were interpolated to a 0.5° grid; the actual ERA5 model resolution is finer than 0.5°.

Yes, thanks.

Line 167: Is "intense" warranted here? What is the quantitative basis for this adjective in this context?

We agree with this suggestion. We now simply write "above a baroclinic zone".

Comment on Figs. 3 and 4: I recommend the following changes to make these plots easier to read and interpret: (1) make the contours and arrows thicker, (2) make the outline of Ukraine thicker and perhaps plot it in a different color to make it more visible, (3) increase the font size for the lat/lon and color bar labels.

We thank the reviewer for pointing our attention to this matter. We have revised all figures and hope that the resolution is now improved.

Line 225: Is the PV anomaly pattern in the summer composite perhaps a reflection of the occurrence of PV cut-offs?

Yes, indeed, high-PV cutoffs over Eastern Europe repeatedly formed, locally changing the static stability, and thus providing the ideal mesoscale environment for the formation of EPEs and triggering cyclogenesis over Ukraine. Also note, that the Black Sea region is characterized by a local maximum in the frequency of PV cutoffs in all seasons (see Fig. 3 in Portmann et al., 2021). We add a brief discussion of this to the text.

Line 235: How much variability is there among the events in the composites with respect to the PV anomaly pattern? It might be worthwhile to also plot the composite standard deviation as a measure of the case-to-case variability.

We have plots for the standard deviation of PV for each season (Fig. R2). The standard deviation exhibits a seasonal variation, reaching its peak (approximately 1,7 SD) during winter and reaching a minimum of 0,7 SD in summer, in the main PV anomaly regions. We note, however, that these fields should be regarded with caution in all seasons except summer, because of the low number of events.

[Figure]

**Fig. R2 Variability of PV anomalies during EPEs for the different seasons (in standard deviation units).**

Line 238: It is not clear to me what the authors mean by "northward", "southward", "eastward", and "westward" here. Do these descriptions refer to the direction of the PV anomaly gradient vector?

Here we mean that in different seasons, the location of PV anomalies in relation to the region of Ukraine varies. In winter, the PV anomaly is situated north (northeast) of Ukraine, in spring it shifts to the west, in summer it moves to the south, and in autumn, it is positioned to the east.

Lines 239–240: I encourage the authors to include a discussion of the possible implications of the composite PV anomaly patterns for forcing of vertical motion over Ukraine. In my opinion, a more direct link needs to be established between the composite flow patterns and the processes that caused the extreme precipitation.

Yes, indeed, we have added the following: "In each season, EPEs appear to be preconditioned largely by a moist southwestern flow at the eastern flank of the upper-level PV anomalies, which leads to dynamic lifting via positive vorticity advection, causing reduced tropospheric stability. Their localization corresponds to the seasonal location of PV anomalies." (L.247-251).

Lines 405–406: When making statements for which direct evidence is not shown, I recommend including "(not shown)."

Added. Thanks!

Lines 463–465: "The exception were winter EPEs…" This conclusion seems inconsistent with the 500-hPa Z and PV anomaly composites for the winter EPEs, which appear to depict strong troughs immediately upstream of Ukraine.

Yes, this conclusion was made for two individual cases (28 December 1999 and 21 December 1993), where a relatively clear zonal upper tropospheric flow over Ukraine could be identified. In the case of 28 December 1999, a shallow short-wave disturbance in the westerly flow was observed. When comparing the composites of Z and PV anomalies with individual cases, we cannot expect to see an identical pattern of the distribution of meteorological parameters. This is because these composites represent maps for all winter cases, and they may not always perfectly coincide with specific synoptic events.

**Typographical corrections**

Line 229: I suggest removing "but not least" here.

Yes, agree.

Line 238: Change "consistently" to "consistent"

Changed. Thanks!

Line 285: Change "more important" to "greater"

Changed. Thanks!

Line 309: Remove "very"

Removed. Thanks!

**References:**

1. Tramblay, Y., Neppel, L., Carreau, J., and Najib, K., Non-stationary frequency analysis of heavy rainfall events in southern France: Hydrological Sciences Journal. 58 (2). 280–294. https://doi.org/10.1080/02626667.2012.754988, 2013.

2. Martin-Vide, J., Sanchez-Lorenzo, A., Lopez-Bustins, J. A., Cordobilla, M. J., Garcia-Manuel, A., and Raso, J. M.: Torrential rainfall in northeast of the Iberian Peninsula: synoptic patterns and WeMO influence, Adv. Sci. Res., 2, 99–105, https://doi.org/10.5194/asr-2-99-2008, 2008.

3. L. Boissier, Freddy Vinet. Paramètres hydroclimatiques et mortalité due aux crues torrentielles : Etude dans le sud de la France. *XXIIème colloque de l'Association Internationale de Climatologie, Cluj-Napoca (Roumanie), 1-5 sept. 2009*, 2009, Cluj-Napoca, Roumanie. pp.79-84. ⟨hal-03069229⟩

4. Alexander L.V.: Global observed long-term changes intemperature and precipitation extremes: A review of progress and limitations in IPCC assessments and beyond. Weather and Climate Extremes, 11 4-16. https://doi.org/10.1016/j.wace.2015.10.007, 2016.

5. Mastrantonas, N., Magnusson, L., Pappenberger, F., Matschullat, J.: Extreme precipitation events in the Mediterranean: Spatiotemporal characteristics and connection to large-scale atmospheric flow patterns, *Quart. J. Roy. Meteor. Soc.,* 148, 875-890, https://doi.org/10.1002/joc.6985, 2020.

6. Hoskins, B., McIntyre, M., and Robertson, A.: On the use and significance of isentropic potential vorticity maps, Q. J. Roy. Meteorol. Soc., 111, 877–946, https://doi.org/10.1256/smsqj.47001, 1985.

7. Portmann, R., M. Sprenger, and H. Wernli: The three-dimensional life cycles of potential vorticity cutoffs: a global and selected regional climatologies in ERA-Interim (1979–2018), Weather Clim. Dynam., 2, 507–534, https://doi.org/10.5194/wcd-2-507-2021, 2021.

---

## Author Comment (AC2)

*Paper egusphere-2023-2594*

**Precipitation extremes in Ukraine from 1979 to 2019: Climatology, large-scale flow conditions, and moisture sources**

by Ellina Agayar, Franziska Aemisegger, Moshe Armon, Alexander Scherrmann, and Heini Wernli

***Final author comments***

We thank all three reviewers for their thoughtful and constructive comments that help us to improve the manuscript. Based on the reviewers' suggestions, we implement several changes in the manuscript. The main changes are that:

- We add a new figure to show more insight into the EPEs in summer (in response to suggestion by reviewer #3).
- We further clarify the data availability (in response to comment by reviewer #3).
- We add more discussion about the moisture source composites in comparison to those for upper-level PV (in response to comment by reviewer #2).
- We better motivate the choice of the 100 mm per day threshold to identify EPEs (in response to comment by reviewers #1 and 2).

Below we provide a one-to-one response to all points raised by the reviewers. The reviewers' comments are in red and our replies in black.

**Reviewer #1**

This paper presents a analysis of EPEs and their climatological drivers in Ukraine using a combination of advanced techniques and models. It identified the common trend of anomalies associated with EPEs in different seasons, and moisture sources of the EPEs. The findings like the important role of land evapotranspiration and the formation of an upper-level trough in all seasons expect winter are quite interesting and informational on the relationships between cyclones and flood related hazards. Overall I find the work to be interesting and sound, and the paper well presented. However, I believe the paper can be better if there are more physical background discussed either in section 3 or section 4. The findings are interesting and I believe that they are important, thus it's very natural for the audience to wonder what could be the physical explanations of the findings (such as geospatial patterns, relationships between the trough and precipitations, the difference between winter and other seasons). I would recommend the authors to have more of these discussed, better with references to former studies, together with the description of their findings.

Overall, I would recommend that this manuscript is suitable for publication in this journal ensuing to the authors addressing the major concern above, and my minor comments below.

We are very grateful to Reviewer #1 for her/his thorough analysis of our work, and for all the comments and suggestions regarding our paper. Incorporating solutions to these particular questions into the revised version of our manuscript will enhance its overall quality.

1.Line 45 – 48: Cite the data source in the corresponding format (newspaper, research paper, book, conference meeting, website, etc.)

Yes, we added a reference (Ukrainian State Agency of Water Resources; Mykhailiuk, 2022).

2. Line 54-55: It would be better if the authors can briefly talk about how the cyclones and blocking systems are relevant to EPEs.

We added a brief description of the mechanism of interaction between blocking systems and cyclones.

"This pattern effectively hinders the usual westerly large-scale atmospheric flow, resulting in flow anomalies around the blocking system and persistent conditions in its immediate region. The mechanism for the formation of any type of blocking circulation involves the nonlinear amplification of atmospheric Rossby waves, ultimately leading to their breaking. Blocks are long-lasting, quasi-stationary systems that frequently occur over specific regions (Moore et al., 2019). Their presence and characteristics significantly impact the occurrence of surface weather extremes (Rex, 1950a; Lenggenhager et al., 2019; Kautz et al., 2022), including EPEs. Furthermore, the most extreme weather is often associated with atmospheric blocking and coexisting upper-tropospheric cutoffs (Portmann et al., 2021)."

3. Line 68: ERA5 is an important data source in this paper, but it was not introduced properly. I would recommend the authors to introduce it before its first appearance in the paper.

Yes, we added: "For this, we use the dataset ERA5, which is the fifth-generation reanalysis from the European Centre for Medium-range Weather Forecasts (ECMWF) that is available since 1940. ERA5 provides hourly estimates for a large number of atmospheric, ocean-wave and land-surface quantities."

4. Line 114: I'm not quite familiar about this so I can be wrong, but I'm wondering if the threshold 0.025 g/(kg*h) is a common practice in this research field. If not, I would recommend the authors to briefly justify their choice or support it with former studies.

For the identification of moisture sources, we apply the methodology by Sodemann et al. (2008). This approach identifies moisture sources from positive increments of specific humidity along a trajectory. We only consider changes in specific humidity exceeding 0.025 g/(kg*h) (detection threshold), because this threshold suppresses spurious uptakes due to numerical noise and keeps the analysis computationally feasible. Currently, this is a frequently used method for determining moisture sources (see, e.g., the references in the paper and Papritz et al., 2021; 2022).

5. Line 238: Add a comma before "southward in summer".

Added. Thanks!

Line 264: The period from the former text paragraph seems to be after the figures and the caption.

Yes, you are right.

7. Line 338, 348, 359, 371, 391, 400: Use the standard citation format of ESWD. The citation needs to be both in text and in the reference list.

Added. Thanks!

**References:**
1. Dotzek, N., P. Groenemeijer, B. Feuerstein, and A. M. Holzer, 2009: Overview of ESSL's severe convective storms research using the European Severe Weather Data-base ESWD. *Atmos. Res*., **93**, 575–86, https://doi.org/10.1016/j.atmosres.2008.10.020.

2. State Agency of Water Resources: https://davr.gov.ua/korotkij-oglyad-potochnoi-vodnoi-situacii-v-richkovih-basejnah-ukraini-stanom-na-26062020

3. Lenggenhager, S., Martius, O.: Atmospheric blocks modulate the odds of heavy precipitation events in Europe, Clim. Dyn., 53, 4155–4171, https://doi.org/10.1007/s00382-019-04779-0, 2019.

4. Moore, B. J., Keyser, D., and Bosart, L. F.: Linkages between extreme precipitation events in the central and eastern United States and Rossby wave breaking, Mon. Wea. Rev., 147, 3327–3349, https://doi.org/10.1175/MWR-D-19-0047.1, 2019

5. Mykhailiuk, R.: Measures to protect the principal Carpathia from disasterable floods by analysis of their causes and consequences in 2008 and 2020. Ecological Safety and Balanced Use of Resources, 2(24), 13–26, https://doi.org/10.31471/2415-3184-2021-2(24)-13-26, 2022.

6. Papritz, L., Aemisegger, F., Wernli, H.: Sources and Transport Pathways of Precipitating Waters in Cold-Season Deep North Atlantic Cyclones, J. Atmos. Sci., 78, 3349-3368, https://doi.org/10.1175/JAS-D-21-0105.s1, 2021.

7. Papritz, L., Hauswirth, D., and Hartmuth, K.: Moisture origin, transport pathways, and driving processes of intense wintertime moisture transport into the Arctic, Weather Clim. Dynam., 3, 1–20, https://doi.org/10.5194/wcd-3-1-2022, 2022.

8. Rex, D.F. : Blocking action in the middle troposphere and its effect upon regional climate: I. An aerological study of blocking action. Tellus 1950a, 2, 196–211.

9. Sodemann, H., Schwierz, C., and Wernli, H.: Interannual variability of Greenland winter precipitation sources: Lagrangian moisture diagnostic and North Atlantic Oscillation influence, J. Geophys. Res.,113, https://doi.org/10.1029/2007JD008503, 2008.

10. Kautz, L.-A., Martius, O., Pfahl, S., Pinto, J., Ramos, A., Sousa, P., and Woollings, T..: Atmospheric Blocking and Weather Extremes over the Euro-Atlantic Sector – A Review. Weather Clim. Dynam., 3, 305–336, https://doi.org/10.5194/wcd-2021-56, 2022.

11. Portmann, R., Sprenger, M., Wernli, H.: The three-dimensional life cycles of potential vorticity cutoffs: a global and selected regional climatologies in ERA-Interim (1979–2018), Weather Clim. Dynam., 2, 507–534, https://doi.org/10.5194/wcd-2-507-2021, 2021.

---

## Author Comment (AC3)

*Paper egusphere-2023-2594*

**Precipitation extremes in Ukraine from 1979 to 2019: Climatology, large-scale flow conditions, and moisture sources**

by Ellina Agayar, Franziska Aemisegger, Moshe Armon, Alexander Scherrmann, and Heini Wernli

***Final author comments***

We thank all three reviewers for their thoughtful and constructive comments that help us to improve the manuscript. Based on the reviewers' suggestions, we implement several changes in the manuscript. The main changes are that:

- We add a new figure to show more insight into the EPEs in summer (in response to suggestion by reviewer #3).
- We further clarify the data availability (in response to comment by reviewer #3).
- We add more discussion about the moisture source composites in comparison to those for upper-level PV (in response to comment by reviewer #2).
- We better motivate the choice of the 100 mm per day threshold to identify EPEs (in response to comment by reviewers #1 and 2).

Below we provide a one-to-one response to all points raised by the reviewers. The reviewers' comments are in red and our replies in black.

**Reviewer #2**

The authors present a study of precipitation extremes in recent decades focussed on Ukraine, based on reanalysis data. Combining dynamical parameters with diagnostic fields for precipitation origin, the authors document commonalities in the atmospheric state between extreme precipitation cases, but also show pronounced variability, in particular with regards to the precipitation sources. Similar information has been provided before for neighboring areas, but not in the region studied here. The paper has a logical structure, is overall well-written, and the figures are in general of good quality. I have a few, mostly minor comments with regard to how the analysis could be strengthened further. These remarks concern the methods, presentation of the results, and their relation to published literature. I hope that these comments will be helpful for the authors in their preparation of a revised manuscript.

We thank Reviewer #2 for the helpful comments and are glad to see the main messages of this manuscript are acknowledged and appreciated by the reviewer.

**Minor comments**

1. The selection of the extreme precipitation events could be better justified. For example, to which percentile does the 100 mm day$^{-1}$ threshold correspond to in different regions of Ukraine? With a constant threshold value, it seems that some events could be more extreme in

some regions than in others. How sensitive are the results to this choice of threshold? This item is also connected to a better presentation of the climatological precipitation pattern in the study region (see below).

We agree that we should better motivate the choice of our threshold, and also, that alternative approaches of identifying extreme events (e.g., with a percentile threshold) could also be meaningful. We added the following: "Our threshold of 100 mm day$^{-1}$, exceeded on average once a year in the stations of the study area, is chosen from expert knowledge, as it is often used to define extreme precipitation events in different countries. For instance, Martin-Vide et al. (2008) used this threshold to determine EPEs in the western Mediterranean, and Tramblay et al. (2013) in southern France. Boissier and Vinet (2009) identified the value of 100 mm day$^{-1}$ as a critical threshold that could trigger fatalities. Also in Ukraine, this threshold is used to identify an event as extreme.

In order to relate this threshold to a percentile we consider eight stations across different regions of Ukraine: two stations in the west: Chernovci and Yaremche; two in the central part: Nejin and Olevsk; two in the eastern part: Izum and Mariupol; and two stations in the south: Odessa and Ai-Petry. For all stations in mountainous areas, as well as in flat regions, the threshold of 100 mm day$^{-1}$ corresponds to the 99.9th percentile or higher. These percentiles highlight that the selected threshold of $\geq 100$ mm day$^{-1}$ indeed selects extreme, i.e., very rare events." These events are so rare that we cannot robustly assess regional differences of percentiles (recall that in total we have only 75 events at all stations).

2. Figure 1 does not provide a lot of information. It could be more informative to instead show for example the seasonal precipitation total in a 4-panel figure, and place the events with their maximum precipitation as text labels on top of the background.

We appreciate the reviewer's comment; indeed, it would be interesting to show the seasonal precipitation total. However, on one hand, the seasonal and annual distribution of precipitation in Ukraine is well presented in the book "Climate of Ukraine" and other atlases (https://uhmi.org.ua/conf/climate_changes/presentation_pdf/plenary_session/Lipinskiy_et_al. pdf, see page 15). Climatological precipitation is largest in the west of Ukraine (annual total >1000 mm), and generally lower along the coast of the Black Sea (annual total <500 mm), except for again high values in the south of Crimea. On the other hand, Fig. 1 clearly shows the geographical features of the territory. If EPE data was superimposed on climatological precipitation maps, this visibility might be compromised. We therefore decided to leave Fig. 1 as is, but we will include a brief description of the precipitation climatology in Ukraine in the revised paper.

3. I find the results do nicely align with several other studies that have been done regarding the moisture sources of extreme precipitation in the Mediterranean and Central Europe, maybe also other regions, that are cited in the introduction. However, I found the discussion a bit brief, and more specific comparison could be done to the existing literature after presentation of the results. For example, the authors find, in agreement with above mentioned previous studies, that there is more structure/regularity in the upper-level circulation than in the moisture sources fields. Why is that so, and what does that imply? At least it could be stated as an overarching finding, and the question be raised, even if the authors do not want to speculate about possible reasons.

We thank Reviewer #2 for this suggestion, and we added the following discussion to Sect. 3.3 (Seasonal mean moisture sources): "It is noteworthy, that there is less structure in the fields of moisture sources compared to the upper-level circulation. This may be due to the fact that the upper-level circulation is often governed by coherent flow features. For example, the presence of a strong jet stream or a well-defined upper-level trough. This implies their somewhat more consistent structure compared to the more variable moisture sources. Since by far most of the global water vapor is located in the lower troposphere, moisture source fields are influenced by factors like sea surface temperatures, local evaporation, soil moisture availability, moisture transport, and low-level winds. For example, Winschall et al., (2014) investigated the importance of intensified local and remote evaporation for Mediterranean precipitation extremes. Krug et al. (2022) determined that the evaporation anomalies are related to wind-speed anomalies indicating mainly dynamically driven evaporation. Grams et al. (2014) emphasized the significant role of soil moisture preconditioning. For instance, intense precipitation events can moisten the previously dry soil and might subsequently serve as moisture sources for subsequent extreme precipitation events (Bohlinger et al., 2017). This complex interaction between various preconditioning factors and the eventually emerging moisture source patterns should be investigated in more detail in future research."

4. I did not find Table 1 so useful, at least not in this location in the paper. Maybe this table should rather be introduced along with the seasonal results? There is also a lot to read in this table, which seems almost like a duplication of the writing in the results section. Maybe the table could be simplified, or some kind of coding of different "event types" could be devised, such that the table provides more comparable information at a glance?

We appreciate the reviewer's comment. Indeed, it is probable that this table would be better placed along with the seasonal results, because it summarizes all these findings. However, we have chosen not to shorten the content in the current version, because we think this table provides a clear and relatively complete overview of seasonal differences, taking into account quantitative changes in the presented parameters. But we (a) introduced some abbreviations for geographical definitions to shorten the text, (b) changed the term "positive PV anomaly" to PV+, and (c) added text in the caption to explain the table better. We hope that the revised version is clearer for the readers and the changes sufficient for the reviewer.

**References:**

1. Tramblay, Y., Neppel, L., Carreau, J., and Najib, K., Non-stationary frequency analysis of heavy rainfall events in southern France: Hydrological Sciences Journal. 58 (2). 280–294. https://doi.org/10.1080/02626667.2012.754988, 2013.

2. Martin-Vide, J., Sanchez-Lorenzo, A., Lopez-Bustins, J. A., Cordobilla, M. J., Garcia-Manuel, A., and Raso, J. M.: Torrential rainfall in northeast of the Iberian Peninsula: synoptic patterns and WeMO influence, Adv. Sci. Res., 2, 99–105, https://doi.org/10.5194/asr-2-99-2008, 2008.

3. Boissier, L., and Vinet, F.: Paramètres hydroclimatiques et mortalité due aux crues torrentielles : Etude dans le sud de la France. *XXIIème colloque de l'Association Internationale de Climatologie, Cluj-Napoca (Roumanie), 1-5 sept. 2009*, 2009, Cluj-Napoca, Roumanie. pp.79-84. ⟨hal-03069229⟩

4. Alexander L.V.: Global observed long-term changes intemperature and precipitation extremes: A review of progress and limitations in IPCC assessments and beyond. Weather and Climate Extremes, 11 4-16. https://doi.org/10.1016/j.wace.2015.10.007, 2016.

5. Mastrantonas, N., Magnusson, L., Pappenberger, F., Matschullat, J.: Extreme precipitation events in the Mediterranean: Spatiotemporal characteristics and connection to large-scale atmospheric flow patterns, *Quart. J. Roy. Meteor. Soc.,* 148, 875-890, https://doi.org/10.1002/joc.6985, 2020.

6. Winschall, A., Sodemann, H., Pfahl S., Wernli H.: How importantis intensified evaporation for Mediterranean precipitation extremes?, J. Geophys. Res. Atmos.,119, 5240–5256, https://doi.org/10.1002/2013JD021175, 2014.

7. Krug, A., Aemisegger, F., Sprenger, M. et al.: Moisture sources of heavy precipitation in Central Europe in synoptic situations with Vb-cyclones, Clim. Dyn., 59, 3227–3245, https://doi.org/10.1007/s00382-022-06256-7, 2022.

8. Grams, C. M., Binder, H., Pfahl, S., Piaget, N., Wernli, H.: Atmospheric processes triggering the central European floods in June 2013, Nat Hazard Earth Syst. Sci., 14, 1691–1702, https://doi.org/10.5194/nhess-14-1691-2014, 2014.

9. Bohlinger, P., Sorteberg, A., Sodemann, H.: Synoptic conditions and moisture sources actuating extreme precipitation in Nepal, J. Geophys. Res. Atmos., 122, 12,653–12,671, https://doi.org/10.1002/2017JD027543, 2017.

---

## Author Response (AR1)

*Paper egusphere-2023-2594*

**Precipitation extremes in Ukraine from 1979 to 2019: Climatology, large-scale flow conditions, and moisture sources**

by Ellina Agayar, Franziska Aemisegger, Moshe Armon, Alexander Scherrmann, and Heini Wernli

*Final author comments*

We thank all three reviewers for their thoughtful and constructive comments that help us to improve the manuscript. Based on the reviewers' suggestions, we implement several changes in the manuscript. The main changes are that:

- We add a new figure to show more insight into the EPEs in summer (in response to suggestion by reviewer #3). Figure 4, P.8.
- We further clarify the data availability (in response to comment by reviewer #3). L102-109
- We add more discussion about the moisture source composites in comparison to those for upper-level PV (in response to comment by reviewer #2). L369-382.
- We better motivate the choice of the 100 mm per day threshold to identify EPEs (in response to comment by reviewers #1 and 2). L110-118.

Below we provide a one-to-one response to all points raised by the reviewers. The reviewers' comments are in red and our replies in black.

**Reviewer #1**

This paper presents a analysis of EPEs and their climatological drivers in Ukraine using a combination of advanced techniques and models. It identified the common trend of anomalies associated with EPEs in different seasons, and moisture sources of the EPEs. The findings like the important role of land evapotranspiration and the formation of an upper-level trough in all seasons expect winter are quite interesting and informational on the relationships between cyclones and flood related hazards. Overall I find the work to be interesting and sound, and the paper well presented. However, I believe the paper can be better if there are more physical background discussed either in section 3 or section 4. The findings are interesting and I believe that they are important, thus it's very natural for the audience to wonder what could be the physical explanations of the findings (such as geospatial patterns, relationships between the trough and precipitations, the difference between winter and other seasons). I would recommend the authors to have more of these discussed, better with references to former studies, together with the description of their findings.

Overall, I would recommend that this manuscript is suitable for publication in this journal ensuing to the authors addressing the major concern above, and my minor comments below.

We are very grateful to Reviewer #1 for her/his thorough analysis of our work, and for all the comments and suggestions regarding our paper. Incorporating solutions to these particular questions into the revised version of our manuscript will enhance its overall quality.

Main comment:

To address the reviewer's main comment about adding physical explanations on the relationships between synoptic features and precipitation as well as more references to former studies we made the following changes:

- We added references to de Vries, 2022 and Pfahl and Wernli, 2012 in the introduction to discuss the relationship between EPEs and different synoptic-scale features such as cyclones and anticyclones in more details (L60-69).
- We added analyses of composites of total precipitation, TCW and CAPE for a more complete picture of the synoptic-scale characteristics of the EPEs in Ukraine to the Sect. 3.2 in Fig. 4 for JJA, and similar figures to the Supplement S3, Fig. S1 for the other seasons (L235-268). We briefly discuss the differences between seasons.
- We added more discussion about the moisture source composites in comparison to those for upper-level PV and provide a concise overview of the factors that explain, why there is less structure in the fields of moisture sources compared to the upper-level circulation fields. (L369-382)

1.Line 45 – 48: Cite the data source in the corresponding format (newspaper, research paper, book, conference meeting, website, etc.)

Yes, we added a reference (Ukrainian State Agency of Water Resources; Mykhailiuk, 2022). L54.

2. Line 54-55: It would be better if the authors can briefly talk about how the cyclones and blocking systems are relevant to EPEs.

We added a brief description of the mechanism of interaction between blocking systems and cyclones. L.62-69.

"It is quite common for heavy precipitation to occur in synoptic configurations at the interface between high-pressure disturbances and cyclones (Breugem et al., 2020). According to Pfahl and Wernli (2012),  in many regions, cyclones are linked with a large percentage of EPEs. Cyclones and anticyclones both play an important role in moisture transport, while cyclones typically also go along with forcing for ascent, in combination leading to EPEs. Blocking anticyclones in addition effectively hinder the usual westerly large-scale atmospheric flow, resulting in persistent flow anomalies in and around the blocked region. Their presence and characteristics significantly impact the predictability of weather extremes (Rex, 1950a; Lenggenhager et al., 2019; Kautz et al., 2022), including EPEs. Furthermore, extreme precipitation is often associated with atmospheric blocking and coexisting upper-tropospheric cutoffs (Portmann et al., 2021)."

3. Line 68: ERA5 is an important data source in this paper, but it was not introduced properly. I would recommend the authors to introduce it before its first appearance in the paper.

Yes, we added: "For this, we use the ERA5 dataset, which is the fifth-generation reanalysis from the European Centre for Medium-range Weather Forecasts (ECMWF) that is available since 1940 (Hersbach et al., 2020). ERA5 provides hourly estimates for a large number of atmospheric, ocean-wave and land-surface quantities." L. 79-82.

4. Line 114: I'm not quite familiar about this so I can be wrong, but I'm wondering if the threshold 0.025 g/(kg*h) is a common practice in this research field. If not, I would recommend the authors to briefly justify their choice or support it with former studies.

For the identification of moisture sources, we apply the methodology by Sodemann et al. (2008). This approach identifies moisture sources from positive increments of specific humidity along a trajectory. We only consider changes in specific humidity exceeding 0.025 g/(kg*h) (detection threshold), because this threshold suppresses spurious uptakes due to numerical noise and keeps the analysis computationally feasible. Currently, this is a frequently used method for determining moisture sources (see, e.g., the references in the paper and Papritz et al., 2021; 2022).

5. Line 238: Add a comma before "southward in summer".

Added. Thanks! L321.

6. Line 264: The period from the former text paragraph seems to be after the figures and the caption.

Yes, you are right. We changed it.

7. Line 338, 348, 359, 371, 391, 400: Use the standard citation format of ESWD. The citation needs to be both in text and in the reference list.

Added. Thanks! L455, L465, L478, L495, L514, L523.

**References:**

1.de Vries, A. J.: A global climatological perspective on the importance of Rossby wave breaking and intense moisture transport for extreme precipitation events, Weather Clim. Dynam., 2, 129–161, https://doi.org/10.5194/wcd-2-129-2021, 2021.
2. State Agency of Water Resources: https://davr.gov.ua/korotkij-oglyad-potochnoi-vodnoi-situacii-v-richkovih-basejnah-ukraini-stanom-na-26062020.
3. Mykhailiuk, R.: Measures to protect the principal Carpathia from disasterable floods by analysis of their causes and consequences in 2008 and 2020, ecological safety and balanced use of resources, 2(24), 13–26, https://doi.org/10.31471/2415-3184-2021-2(24)-13-26, 2022.
4.Pfahl, S. and Wernli, H.: Quantifying the relevance of cyclones for precipitation extremes, J. Climate, 25, 6770–6780, https://doi.org/10.3929/ethz-b-000057303, 2012a.
5.Breugem A. J., Wesseling, J. G., Oostindie, K. and Ritsema, C. J.: Meteorological aspects of heavy precipitation in relation to floods – An overview, Earth-Science Reviews, V.204, 103171, ISSN 0012-8252, https://doi.org/10.1016/j.earscirev.2020.103171, 2020.
6. Rex, D.F. : Blocking action in the middle troposphere and its effect upon regional climate: I. An aerological study of blocking action. Tellus, 2, 196–211. https://doi.org/10.3402/tellusa.v2i3.8546, 1950.

7. Lenggenhager, S., Martius, O.: Atmospheric blocks modulate the odds of heavy precipitation events in Europe, Clim. Dyn., 53, 4155–4171, https://doi.org/10.1007/s00382-019-04779-0, 2019.

8. Kautz, L.-A., Martius, O., Pfahl, S., Pinto, J., Ramos, A., Sousa, P., and Woollings, T.: Atmospheric blocking and weather extremes over the Euro-Atlantic sector, – A Review. Weather Clim. Dynam., 3, 305–336, https://doi.org/10.5194/wcd-2021-56, 2022.

9. Portmann, R., Sprenger, M., and Wernli, H.: The three-dimensional life cycles of potential vorticity cutoffs: a global and selected regional climatologies in ERA-Interim (1979–2018), Weather Clim. Dynam., 2, 507–534, https://doi.org/10.5194/wcd-2-507-2021, 2021.

10. Hersbach, H., and Coauthors: The ERA5 global reanalysis. Quart. J. Roy. Meteor. Soc., 146, 1999–2049, https://doi.org/10.1002/qj.3803, 2020.

11. Sodemann, H., Schwierz, C., and Wernli, H.: Interannual variability of Greenland winter precipitation sources: Lagrangian moisture diagnostic and North Atlantic Oscillation influence, J. Geophys. Res.,113, https://doi.org/10.1029/2007JD008503, 2008.

12. Papritz, L., Aemisegger, F., and Wernli, H.: Sources and Transport Pathways of Precipitating Waters in Cold-Season Deep North Atlantic Cyclones, J. Atmos. Sci., 78, 3349-3368, https://doi.org/10.1175/JAS-D-21-0105.s1, 2021.

13. Papritz, L., Hauswirth, D., and Hartmuth, K.: Moisture origin, transport pathways, and driving processes of intense wintertime moisture transport into the Arctic, Weather Clim. Dynam., 3, 1–20, https://doi.org/10.5194/wcd-3-1-2022, 2022.

14. Dotzek, N., P. Groenemeijer, B. Feuerstein, and Holzer, A. M.: Overview of ESSL's severe convective storms research using the European Severe Weather Data-base ESWD. Atmos. Res., 93, 575–86, https://doi.org/10.1016/j.atmosres.2008.10.020, 2009.

**Reviewer #2**

The authors present a study of precipitation extremes in recent decades focussed on Ukraine, based on reanalysis data. Combining dynamical parameters with diagnostic fields for precipitation origin, the authors document commonalities in the atmospheric state between extreme precipitation cases, but also show pronounced variability, in particular with regards to the precipitation sources. Similar information has been provided before for neighboring areas, but not in the region studied here. The paper has a logical structure, is overall well-written, and the figures are in general of good quality. I have a few, mostly minor comments with regard to how the analysis could be strengthened further. These remarks concern the methods, presentation of the results, and their relation to published literature. I hope that these comments will be helpful for the authors in their preparation of a revised manuscript.

We thank Reviewer #2 for the helpful comments and are glad to see the main messages of this manuscript are acknowledged and appreciated by the reviewer.

**Minor comments**

1. The selection of the extreme precipitation events could be better justified. For example, to which percentile does the 100 mm day$^{-1}$ threshold correspond to in different regions of Ukraine? With a constant threshold value, it seems that some events could be more extreme in some regions than in others. How sensitive are the results to this choice of threshold? This item is also connected to a better presentation of the climatological precipitation pattern in the study region (see below).

We agree that we should better motivate the choice of our threshold, and also, that alternative approaches of identifying extreme events (e.g., with a percentile threshold) could also be meaningful. However, we do believe the 100 mm day$^{-1}$ is a relevant threshold. Therefore, we added the following: "Our criterion to identify EPEs was a threshold of 100 mm day$^{-1}$. With this criterion, in total 82 EPEs were identified. Table S1 in the Supplement lists the date and station for each of these events. Our threshold of 100 mm day$^{-1}$, is chosen from expert knowledge, as it is often used to define EPEs in different countries. For instance, Martin-Vide et al. (2008) used this threshold to determine EPEs in the western Mediterranean, and Tramblay et al. (2013) in southern France. Boissier and Vinet (2009) identified the value of 100 mm day$^{-1}$ as a critical threshold that could trigger fatalities. Also in Ukraine, this threshold is used to identify an event as extreme. Given that we considered a 40 year time period and that EPEs were identified at each station between 0 and 3 times (see Table S1), we can estimate that our threshold corresponds to the 99.8th percentile or higher. These percentiles highlight that the selected threshold of $\geq 100$ mm day$^{-1}$ indeed selects extreme, i.e., very rare events. These events are so rare that we cannot robustly assess regional differences of percentiles". L110-118.

2. Figure 1 does not provide a lot of information. It could be more informative to instead show for example the seasonal precipitation total in a 4-panel figure, and place the events with their maximum precipitation as text labels on top of the background.

We appreciate the reviewer's comment; indeed, it would be interesting to show the seasonal precipitation total. However, on one hand, the seasonal and annual distribution of precipitation in Ukraine is well presented in the book "Climate of Ukraine" and other atlases (https://uhmi.org.ua/conf/climate_changes/presentation_pdf/plenary_session/Lipinskiy_et_al. pdf, see page 15). Precipitation is largest in the west of Ukraine (annual total >1000 mm), and generally lower along the coast of the Black Sea (annual total <500 mm), except for again high values in the south of Crimea. On the other hand, Fig. 1 clearly shows the geographical features of the territory. If climatological data was superimposed on extreme precipitation maps, this visibility might be compromised. We therefore decided to leave Fig. 1 as is, but we will include a brief description of the precipitation climatology in Ukraine in the revised paper: "Precipitation in Ukraine generally exhibits a diminishing trend from the north and northwest to the south and southeast areas (Lipinskyi et al., 2011). In the mountainous areas, orographic lifting contributes to enhanced precipitation. As a result, the Ukrainian Carpathians and the Crimean Mountains experience the largest precipitation (annual total >1000 mm). In the central and eastern parts of Ukraine, the amount of annual precipitation is 550 - 650 mm; the southern part, along the coast of the Black Sea, is comparatively dry (annual total 380 – 400 mm). In the cold season, approximately 20–25% of the annual precipitation occurs, contrasting with the warm period, where 75–80% of the total annual precipitation is recorded. During the warmer season, the precipitation distribution reflects the annual pattern, with a gradual decrease from the northwest to the southeast, reaching 300 mm or less on the coastal regions". L.161-168.

3. I find the results do nicely align with several other studies that have been done regarding the moisture sources of extreme precipitation in the Mediterranean and Central Europe, maybe also other regions, that are cited in the introduction. However, I found the discussion a bit brief, and more specific comparison could be done to the existing literature after presentation of the results. For example, the authors find, in agreement with above mentioned previous studies, that there is more structure/regularity in the upper-level circulation than in

the moisture sources fields. Why is that so, and what does that imply? At least it could be stated as an overarching finding, and the question be raised, even if the authors do not want to speculate about possible reasons.

We thank Reviewer #2 for this suggestion, and we added the following discussion to Sect. 3.4 (Seasonal mean moisture sources): "It is noteworthy, that there is less coherent structure in the fields of moisture sources compared to the upper-level circulation fields investigated in the previous sections. This may be due to the fact that the upper-level circulation is often governed by large-scale flow features, for example, the presence of a strong jet stream or a well-defined upper-level trough. This can explain their somewhat more consistent structure compared to the more variable moisture sources. Since by far most of the global water vapor is located in the lower troposphere, moisture source fields are influenced by factors like sea surface temperatures, local evaporation, soil moisture availability, moisture transport, and low-level winds, and convection. Winschall et al., (2014) investigated the importance of intensified local and remote evaporation for Mediterranean precipitation extremes. Krug et al. (2022) determined that the evaporation anomalies are related to wind-speed anomalies indicating mainly dynamically driven evaporation. Grams et al. (2014) emphasized the significant role of soil moisture preconditioning. For instance, intense precipitation events can moisten the previously dry soil and might subsequently serve as moisture sources for subsequent extreme EPEs (Bohlinger et al., 2017). And lastly, Dahinden et al. (2023) studied shallow and deep convective systems that occur in random patches and lead to highly variable structure to the moisture source maps. This complex interaction between various preconditioning factors and the eventually emerging moisture source patterns should be investigated in more detail in future research " L. 369-382.

4. I did not find Table 1 so useful, at least not in this location in the paper. Maybe this table should rather be introduced along with the seasonal results? There is also a lot to read in this table, which seems almost like a duplication of the writing in the results section. Maybe the table could be simplified, or some kind of coding of different "event types" could be devised, such that the table provides more comparable information at a glance?

We appreciate the reviewer's comment. Indeed, it is probable that this table would be better placed along with the seasonal results, because it summarizes all these findings. However, we have chosen not to shorten the content in the current version, because we think this table provides a clear and relatively complete overview of seasonal differences, taking into account quantitative changes in the presented parameters. But we (a) introduced some abbreviations for geographical definitions to shorten the text, (b) changed the term "positive PV anomaly" to PV+, and (c) added text in the caption to explain the table better (Table 1 Mean seasonal characteristics of EPEs in Ukraine: Anomalies (units), moisture sources (MS, %) and affected regions. Table abbreviations: W-west, E-east, S-south, N-north. We hope that the revised version is clearer for the readers and the changes sufficient for the reviewer. L.391-397.

**References:**

1. Martin-Vide, J., Sanchez-Lorenzo, A., Lopez-Bustins, J. A., Cordobilla, M. J., Garcia-Manuel, A., and Raso, J. M.: Torrential rainfall in northeast of the Iberian Peninsula: synoptic patterns and WeMO influence, Adv. Sci. Res., 2, 99–105, https://doi.org/10.5194/asr-2-99-2008, 2008.

2. Tramblay, Y., Neppel, L., Carreau, J., and Najib, K.: Non-stationary frequency analysis of heavy rainfall events in southern France, Hydrol. Scien. Journ., 58, 280–294,https://doi.org/10.1080/02626667.2012.754988, 2013.

3. Boissier, L., and Vinet, F.: Paramètres hydroclimatiques et mortalité due aux crues torrentielles : Etude dans le sud de la France. XXIIème colloque de l'Association Internationale de Climatologie, 1-5 Sept. 2009, Cluj-Napoca, Roumanie. pp.79-84. ⟨hal-03069229⟩, 2009.

4. Lipinskyi, V.M., Osadchy, V., Shestopalov, V.M., Rudenko, L.G., Dmytrenko, V.P., Martazinova, V.F., Nabivanets, Y.B., Babichenko, V.N., Kulbida, N.Y., and Shereshevsky, A.I.: Atlas "Climate and water resources of Ukraine", K.: Nika Center, https://uhmi.org.ua/conf/climate_changes/presentation_pdf/plenary_session/Lipinskiy_et_al.pdf, 2011.

5. Dahinden, F., Aemisegger, F., Wernli, H., and Pfahl, S.: Unravelling the transport of moisture into the Saharan Air Layer using passive tracers and isotopes, Atmospheric Science Letters, 24, e1187, https://doi.org/10.1002/asl2.1187, 2023.

6. Winschall, A., Sodemann, H., Pfahl S., and Wernli H.: How importantis intensified evaporation for Mediterranean precipitation extremes?, J. Geophys. Res. Atmos.,119, 5240–5256, https://doi.org/10.1002/2013JD021175, 2014.

7. Krug, A., Aemisegger, F., Sprenger, M., and Ahrens, B.: Moisture sources of heavy precipitation in Central Europe in synoptic situations with Vb-cyclones, Clim. Dyn., 59, 3227–3245, https://doi.org/10.1007/s00382-022-06256-7, 2022.

8. Grams, C. M., Binder, H., Pfahl, S., Piaget, N., and Wernli, H.: Atmospheric processes triggering the central European floods in June 2013, Nat Hazard Earth Syst. Sci., 14, 1691–1702, https://doi.org/10.5194/nhess-14-1691-2014, 2014.

9. Bohlinger, P., Sorteberg, A., and Sodemann, H.: Synoptic conditions and moisture sources actuating extreme precipitation in Nepal, J. Geophys. Res. Atmos., 122, 12,653–12,671, https://doi.org/10.1002/2017JD027543, 2017.

**Reviewer #3**

Overall, I found this manuscript to be clear, concise, and well-written. The study provides a novel climatological investigation of extreme precipitation events (EPEs) in Ukraine, documenting the synoptic-scale conditions in which these events occur and quantifying the moisture sources using a Lagrangian trajectory-based diagnostic. The findings help to address a gap in scientific understanding regarding EPEs in Ukraine.

While the paper is strong overall, I have a number of comments for the authors to consider. Once these comments are satisfactorily addressed, the manuscript may be acceptable for publication.

We express our sincere gratitude to Reviewer #3 for their comprehensive analysis of our work and for providing valuable comments and suggestions for our paper. We are confident that integrating solutions to these specific questions into the revised version of our manuscript will significantly improve its overall quality.

**Major comments**

In my opinion, the manuscript is lacking in diagnostic analysis of the ingredients and processes resulting in extreme precipitation. Composite analyses and case studies are

presented, and the circulation patterns are discussed, but it is still not entirely clear to me how the ingredients for heavy precipitation are established and maintained for these events. Are these events characterized by, for example, particularly anomalous moisture content or strong ascent? Do the key flow features tend to be particularly slow-moving? The study could be strengthened in this regard by inclusion of additional statistical/composite/case study analyses of key ingredients, such as dynamical forcing for ascent (e.g., quasi-geostrophic forcing or frontogenesis), moisture content/moisture flux, and convective instability (e.g., convective available potential energy). Such analyses would help to elucidate how the circulation features shown in Figs. 3 and 4 are linked to the ingredients for heavy precipitation in Ukraine, thereby providing a more complete picture of the synoptic-scale characteristics of the EPEs. Precipitation composites based on the ERA5 could also help to show where in the region the precipitation tends to be focused for the different seasonal groups of EPEs, thereby providing helpful context when interpreting the composite patterns.

We agree with the reviewer that a more in-depth analysis is useful, and we added analyses of composites of total precipitation, TCW and CAPE for a more complete picture of the synoptic-scale characteristics of the EPEs in Ukraine to the Sect. 3.2. The figure below (Fig. R1; similar to Fig. 4) shows anomalies of the total precipitation, TCW and CAPE in physical units at 15 UTC for EPE days in summer. (However, to avoid overloading the paper with additional figures, we added similar figures to the Supplement for the other seasons). The figure shows anomalously high values of TCW and CAPE over the entire Ukraine on summer EPE days. L235-268.

[Figure]

**Fig. R1 Anomalies of total precipitation, TCW and CAPE, at 15 UTC on EPE days in summer.**

How was the 100 mm day$^{-1}$ threshold selected, and how extreme is it for the various stations? I recommend quantifying where this threshold fits in the climatological distribution at each station. Would it be possible to identify EPEs as daily precipitation totals exceeding an upper percentile (e.g., 95th percentile) of the climatological distribution for each station instead of a fixed threshold?

See our response to minor comment 1 of reviewer #2. Note that with our choice of the threshold we identify less than 100 EPEs at more than 180 stations over 40 years. Therefore, the corresponding percentile is much larger than the 95$^{th}$ percentile mentioned by the reviewer, indicating that this study focuses on much more extreme and very rare events. L110-118.

There is redundancy in showing maps of both 500-hPa geopotential height and near-tropopause PV. Both fields depict qualitatively similar structures and patterns of the upper-level flow. Is it necessary to show both fields?

We agree that there is some redundancy between these fields, but as discussed in many studies about the usefulness of upper-level PV charts (e.g., Hoskins et al., 1985), the PV charts show more structural details that can be useful to understand the dynamics. So far, studies showing PV charts have not been conducted for the Ukrainian domain. On the other hand, the choice of the classical 500-hPa charts is motivated by the fact that it has been widely use in synoptic analyses. Thus, to show the similarities and differences of the two fields, we think that both fields are quite useful for our paper.

**Minor comments**

Line 82: Do all of the stations have the same record length? Are they all available for 1979–2019?

Thank you for this question, it is indeed worth covering this point in more detail. We therefore changed the text as follows:

"For this study, 215 meteorological stations and posts (including aviation weather stations, gauging stations, etc.) with daily data from 1979 to 2019 are used. From this dataset, 183 stations were selected for our study that have a complete set of data. The remaining 32 stations did not have the same record length for various reasons. Nevertheless, these stations were still tested for the occurrence of EPEs, but no extreme events were found according to our criteria (see below). Due to the absence of data in the Ukrainian meteorological network for certain regions of Crimea from February 2015 to December 2019, additional data were obtained using open-access observations for this region (SYNOP observational data). Unfortunately, data for four stations in the Donetsk and Lugansk regions for the period of 2015-2019 are not openly available. In this region, a 36-year dataset was employed to identify days with extreme precipitation." L102-109.

Line 88: It would be more accurate to state that the reanalysis data were interpolated to a 0.5° grid; the actual ERA5 model resolution is finer than 0.5°.

Yes, thanks. L125-126.

Line 167: Is "intense" warranted here? What is the quantitative basis for this adjective in this context?

We agree with this suggestion. We now simply write "above a baroclinic zone". L212.

Comment on Figs. 3 and 4: I recommend the following changes to make these plots easier to read and interpret: (1) make the contours and arrows thicker, (2) make the outline of Ukraine thicker and perhaps plot it in a different color to make it more visible, (3) increase the font size for the lat/lon and color bar labels.

We thank the reviewer for pointing our attention to this matter. We have revised all figures and hope that the resolution is now improved. P. 7,9.

Line 225: Is the PV anomaly pattern in the summer composite perhaps a reflection of the occurrence of PV cut-offs?

Yes, indeed, high-PV cutoffs over Eastern Europe repeatedly formed, locally changing the static stability, and thus providing the ideal mesoscale environment for the formation of EPEs and triggering cyclogenesis over Ukraine. Also note, that the Black Sea region is characterized by a local maximum in the frequency of PV cutoffs in all seasons (see Fig. 3 in Portmann et al., 2021). We add a brief discussion of this to the text. L304-307.

Line 235: How much variability is there among the events in the composites with respect to the PV anomaly pattern? It might be worthwhile to also plot the composite standard deviation as a measure of the case-to-case variability.

We have plots for the PV anomaly in standard deviation units for each season (Fig. R2, similar to Fig. S.2 in the Supplement S3), which makes it possible to overcome differences in PV variance across regions and seasons. The anomaly exhibits a seasonal variation, reaching its peak (approximately 1,7 SD anomaly) during winter and reaching a minimum of 0,7 SD in summer, in the main PV anomaly regions. We note, however, that these fields should be regarded with caution in all seasons except summer, because of the low number of events. L315-317.

[Figure]

**Fig. R2  Variability of PV anomalies during EPEs for the different seasons (in standard deviation units).**

Line 238: It is not clear to me what the authors mean by "northward", "southward", "eastward", and "westward" here. Do these descriptions refer to the direction of the PV anomaly gradient vector?

Here we mean that in different seasons, the location of PV anomalies in relation to the region of Ukraine varies. In winter, the PV anomaly is situated north (northeast) of Ukraine, in spring it shifts to the west, in summer it moves to the south, and in autumn, it is positioned to the east.

Lines 239–240: I encourage the authors to include a discussion of the possible implications of the composite PV anomaly patterns for forcing of vertical motion over Ukraine. In my opinion, a more direct link needs to be established between the composite flow patterns and the processes that caused the extreme precipitation.

Yes, indeed, we have added the following: "In each season, EPEs appear to be preconditioned largely by a moist flow from the southwest, south, or southeast, along the eastern flank of the upper-level PV anomalies." L.323-325.

Lines 405–406: When making statements for which direct evidence is not shown, I recommend including "(not shown)."

Added. Thanks! L529.

Lines 463–465: "The exception were winter EPEs…" This conclusion seems inconsistent with the 500-hPa Z and PV anomaly composites for the winter EPEs, which appear to depict strong troughs immediately upstream of Ukraine.

Yes, this conclusion was made for two individual cases (28 December 1999 and 21 December 1993), where a relatively clear zonal upper tropospheric flow over Ukraine could be identified. In the case of 28 December 1999, a shallow short-wave disturbance in the westerly flow was observed. When comparing the composites of Z and PV anomalies with individual cases, we cannot expect to see an identical pattern of the distribution of meteorological parameters. This is because these composites represent maps for all winter cases, and they may not always perfectly coincide with specific synoptic events.

**Typographical corrections**

Line 229: I suggest removing "but not least" here.

Yes, agree.

Line 238: Change "consistently" to "consistent"

Changed. Thanks! L321.

Line 285: Change "more important" to "greater"

Changed. Thanks! L389.

Line 309: Remove "very"

Removed. Thanks!

**References:**

1. Hoskins, B., McIntyre, M., and Robertson, A.: On the use and significance of isentropic potential vorticity maps, Q. J. Roy. Meteorol. Soc., 111, 877–946, https://doi.org/10.1256/smsqj.47001, 1985.
2. Portmann, R., M. Sprenger, and H. Wernli: The three-dimensional life cycles of potential vorticity cutoffs: a global and selected regional climatologies in ERA-Interim (1979–2018), Weather Clim. Dynam., 2, 507–534, https://doi.org/10.5194/wcd-2-507-2021, 2021.

---

## Author Response (AR2)

*Paper egusphere-2023-2594*

**Precipitation extremes in Ukraine from 1979 to 2019: Climatology, large-scale flow conditions, and moisture sources**

by Ellina Agayar, Franziska Aemisegger, Moshe Armon, Alexander Scherrmann, and Heini Wernli

**Comment from Reviewer and Editor**

"In my previous review, I have made suggestions to revise and improve the presentation of the material in the paper. The authors chose to address these points in a minimal way, for example by adding writing rather than changing Fig. 1. That will probably work, but I feel my comments have not really been taken up in a positive spirit by the authors, and just been dealt with in a minimal way. Same regarding my comments on Table 1. To save also my time, I will therefore not provide further helpful input on this manuscript.

My comments have been addressed, but in a somewhat minimalistic way. For example, Fig. 1 is still not very informative and low quality, and Table 1 still is cluttered and provides too much information that is not well organised. Nonetheless, the paper will probably work that way, even though the presentation quality is not optimal."

We are grateful to the Reviewer for their valuable input on this manuscript and sincerely appreciate reviewer's time and comments. Therefore, we added maps of the seasonal precipitation (to Fig.1 L.173-178), and shifted Table 1 to the Supplementary Material (Table S3).

We hope that the Editor is fine with these changes.